



# Can machine learning correct microwave humidity radiances for the influence of clouds?

Inderpreet Kaur[1], Patrick Eriksson[1], Simon Pfreundschuh[1], and David Ian Duncan[2]

[1]Department of Space, Earth and Environment, Chalmers University of Technology, Gothenburg, Sweden
[2]European Centre for Medium Range Weather Forecasts, Reading, United Kingdom

**Correspondence:** Inderpreet Kaur <kauri@chalmers.se>

**Abstract.**

A methodology based on quantile regression neural networks (QRNN) is presented that identifies and corrects the cloud impact on microwave humidity sounder radiances at 183 GHz. This approach estimates the posterior distributions of noise free clear-sky (NFCS) radiances, providing nearly bias-free estimates of clear-sky radiances with a full posterior error distribution. It is first demonstrated by application to a present sensor, the MicroWave Humidity Sounder-2 (MWHS-2), then the applicability to sub-millimeter (sub-mm) sensors is also analysed. The QRNN results improve upon what operational cloud filtering techniques like a scattering index can achieve, but are ultimately imperfect due to limited information content on cirrus impact from traditional microwave channels—the negative departures associated with high cloud impact are successfully corrected, but thin cirrus clouds cannot be fully corrected. In contrast, when sub-mm observations are used, QRNN successfully corrects most cases with cloud impact, with only 2–6% of the cases left partially corrected. The methodology works well even if only one sub-mm channel (325 GHz) is available. When using sub-mm observations, cloud correction usually results in error distributions with standard deviation less than typical channel noise values. Furthermore, QRNN outputs predicted quantiles for case-specific uncertainty estimates, successfully representing the uncertainty of cloud correction for each observation individually. In comparison to deterministic correction or filtering approaches, the corrected radiances and attendant uncertainty estimates have great potential to be used efficiently in assimilation systems due to being largely unbiased and adding little further uncertainty to the measurements.

*Copyright statement.* TEXT

## 1 Introduction

Satellite observations of humidity inside the troposphere are mainly performed by downward-looking sensors. Among this class of observations, the frequency range around 183 GHz has a special position. Water vapour has a noticeable transition at 22 GHz, but it is relatively weak and only column values can be derived (e.g., Schluessel and Emery, 1990) for the observation geometry of concern. The first transition in the microwave region that can be used to derive altitude information, i.e. "sounding", is the



one at 183 GHz (Kakar, 1983; Wang et al., 1983). On the other hand, at infrared wavelengths a high number of water vapour transitions are found, including some of high strength. As a consequence, infrared sounders can provide humidity profiles with

high precision and good vertical resolution, but with strong limitations imposed by clouds. To be able to also sense humidity inside and below clouds, weather satellites are since some time equipped with channels around 183 GHz. Today such channels are part of several sensors, such as ATMS (Advanced Technology Microwave Sounder, Weng et al., 2012).

Although microwave channels are less affected by cloud contamination, precipitation and most dense clouds, particularly if found at high altitude, can still affect measured radiances around 183 GHz (e.g. Bennartz and Bauer, 2003). As the impact

from the hydrometeors then is dominated by scattering, the complexity of the analysis of the data increases dramatically and there exists a need to identify the problematic cases. This is normally denoted as cloud filtering, to obtain data of "clear sky" character. Such filtering has been applied to derive climate records (Lang et al., 2020) and is essential in studies of the agreement between observations and simulations (Brogniez et al., 2016) as well as comparing observations of different instruments to validate their calibration (John et al., 2013; Moradi et al., 2015; Berg et al., 2016). Commonly used cloud filtering methods

for these applications are based on 183 GHz data alone, involving rules on the brightness temperature differences between channels (Burns et al., 1997; Buehler et al., 2007).

Another motivation necessitating the need for cloud filtering is usage of 183 GHz channels in numerical weather prediction (NWP). Usage of passive microwave data by all-sky assimilation in global NWP is growing (Geer et al., 2017), but 183 GHz data are still mainly used in a clear-sky fashion (Geer et al., 2018). The latter is particularly true in NWP of regional scope

(Gustafsson et al., 2018), with clear-sky assimilation of 183 GHz radiances still commonplace. Regardless, both clear-sky and all-sky assimilation require identification of cloud affected observations, either to screen out these observations or to assign an appropriate observation error. The most commonly used cloud filtering techniques are the "Scattering index" (Geer et al., 2014) and the "observation minus background" (O−B). The first one is based on brightness temperature differences between 89 and 150 GHz. In the second one, the forecast model is used to obtain an estimate of the expected clear-sky value and the

observation is rejected if the deviation exceeds some threshold (English et al., 1999).

At 183 GHz, the impact of hydrometeors typically causes a decrease in the observed radiance due to scattering from ice hydrometeors (e.g., Barlakas and Eriksson, 2020). This implies that if any cloud contamination is missed by the filtering, a negative bias in the mean radiance, compared to the true clear-sky mean, may translate into a bias in humidity after the retrieval or assimilation. For NWP systems assimilating clear-sky observations, the effect of undetected clouds may be overcome by

inflating the observational errors and diminishing the impact of observations. Furthermore, the mathematical assumptions of data assimilation (DA) are predicated Gaussian errors with no mean bias, and residual cloud impacts that cause a net bias are not easily handled by variational bias correction. One solution is to apply a very strict filtering, but this increases the rejection of clear-sky values, i.e. an important loss of useful data. Another limitation of existing filtering approaches is their "one for all" approach, i.e. observations in all 183 GHz channels are either kept or rejected. This often rejects more observations than

needed, as the channels differ in their altitude coverage. An observation could be cloudy in some channels and still the be clear-sky in others. To allow a channel specific filtering, data likely need to be combined in a more complex manner than simple differences, but it is unclear what type of regression would be best as the ideal solution would be scene-dependent.





This points towards applying machine learning techniques (e.g., Favrichon et al., 2019). A maybe less obvious problem is the assignment of uncertainty to the filtered values. To our best knowledge, so far only estimates of mean and worst case errors exist in the literature. Some cases with relatively high cloud impact will likely be missed, while most cases are clear-sky from the start. As the remaining cloudy cases can cause significant biases, the likely solution is to apply a quite conservative (high) error estimate. However, this will unnecessarily downgrade the value of the truly clear-sky cases and the observations are used in a non-optimal manner.

In this study, we approach the cloud filtering task from a new angle. The basic idea is to derive an estimate of the corresponding noise-free clear-sky (NFCS) value (i.e. the radiance that would have been measured in absence of noise and hydrometeors). This is done for each channel separately, only using measurements (no "background" data involved). Not only a best estimate is provided, but also a case-specific uncertainty. This information could be used as a pure filter, by rejecting data where the correction exceeds some threshold value. However, even better is to replace the original value with the predicted NFCS value when forming the clear-sky dataset. We denote this approach as cloud correction. It is shown below that a basically bias free cloud correction can be obtained. This feature also removes the need for defining threshold values, as long as the retrieval or assimilation system can incorporate the uncertainty of the corrected value. As also will be shown, the uncertainty for originally clear-sky data is determined by noise, but the uncertainty increases with magnitude of correction. Accordingly, the cloud correction approach permits the full weight of clear-sky data to be preserved.

The proposed cloud correction scheme makes use of a Quantile Regression Neural Network (QRNN, Pfreundschuh et al., 2018) to obtain a probabilistic prediction of the NFCS value. Unlike traditional neural networks techniques, which typically only provide a point estimate of the target variable, QRNNs are trained to predict an arbitrary set of quantiles of its Bayesian a posteriori distribution (Pfreundschuh et al., 2018). The predicted a posteriori distribution can then be used to derive an estimate of the NFCS value together with an estimate of the corresponding uncertainty.

The main focus of this study is the potential of this cloud-correction method using sub-millimetre (sub-mm) observations, which will become available operationally with the launch of the Ice Cloud Imager (ICI, Eriksson et al., 2020) on board the next generation of European Organisation for the Exploitation of Meteorological Satellites Polar System - Second Generation (EUMETSAT EPS-SG). Additionally, we demonstrate the feasibility of the approach based on 89 and 150 GHz channels (following Geer et al., 2014), which are available on several sensors extant today. The focus on sub-mm channels is motivated by several reasons. First, the higher frequencies are more sensitive to scattering effects from smaller hydrometeors and are thus expected to provide greater sensitivity to high altitude cirrus clouds. For example, in some cloudy situations the cloud impact at 183 GHz may be of the order of thermal noise and modelling uncertainties, while the impact at 325 GHz is significant enough to provide sufficient signal to noise for identifying cloud. Second, the proposed cloud correction methods allow integration of ICI sub-mm observations in clear-sky DA schemes with no further modifications, thus providing a simple way to make use this novel data source as soon as it becomes available.

A description of the data used in this study and the QRNN approach is provided in Sect. 2. In Sect. 3 we demonstrate the applicability of correction scheme to existing sensors, and later its application is extended to include sub-mm channels (Sect. 4). The results are discussed in Sect. 5, and Sect. 6 presents the conclusions from this work, and the future outlook.



**Table 1.** Specifications of MWHS-2 channels relevant to this study.

| Channel | Frequency [GHz] | Bandwidth [MHz] | NEΔT [K] |
|---|---|---|---|
| 1 | 89.0 | 1500 | 1.0 |
| 6 | 118.75±1.1 | 200 | 1.6 |
| 7 | 118.75±2.5 | 200 | 1.6 |
| 10 | 150.0 | 1500 | 1.0 |
| 11 | 183.31±1.0 | 500 | 1.0 |
| 12 | 183.31±1.8 | 700 | 1.0 |
| 13 | 183.31±3.0 | 1000 | 1.0 |
| 14 | 183.31±4.5 | 2000 | 1.0 |
| 15 | 183.31±7.0 | 2000 | 1.0 |

## 2 Data and methods

### 2.1 Satellite Instruments

#### 2.1.1 MicroWave Humidity Sounder-2

The MicroWave Humidity Sounder 2 (MWHS-2) is an instrument on two current satellites in the FengYun-3 series, FY-3C and FY-3D. MWHS-2 is a cross track scanning microwave radiometer and measures 15 frequencies in the range 89–191 GHz. 89 GHz and 150 GHz are window channels, five humidity sounding channels are centered around 183 GHz, and eight temperature sounding channels are centered on the 118 GHz oxygen absorption line. The five humidity sounding channels are similar to ATMS. Observations from MWHS-2 are routinely assimilated in all-sky conditions at the European Centre for Medium-Range Weather Forecasts (ECMWF) with demonstrable positive impact on forecast performance (Duncan and Bormann, 2020). The channels relevant to this study are described in Table 1.

For this study, MWHS-2 simulations were sourced from the operational ECMWF assimilation system.

#### 2.1.2 Ice Cloud Imager

The ICI is a new instrument on board EPS-SG satellite MetOp-SG (Meteorological Operational - Second Generation). MetOp-SG is scheduled for launch in 2024, and it will make ICI the first operational sensor observing Earth using sub-mm wavelengths. The main objective of ICI is to use high frequency channels for measuring ice cloud properties, and improve the representation of ice clouds in regional and global NWP models. ICI is a conically scanning radiometer that will measure 13 frequencies from 183 GHz up to 664 GHz. Among all available channels, 183 GHz, 325 GHz and 448 GHz, will measure vertical polarization; 110 while other channels around 243 GHz and 664 GHz are "window channels" and will measure both vertical and horizontal po-





**Table 2.** Specifications of ICI channels relevant to this study.

| Channel | Frequency [GHz] | Bandwidth [MHz] | NEΔT [K] |
|---------|-----------------|-----------------|----------|
| I1V | 183.31±7.0 | 2000 | 0.8 |
| I2V | 183.31±3.4 | 1500 | 0.8 |
| I3V | 183.31±2.0 | 1500 | 0.8 |
| I5V | 325.15±9.5 | 3000 | 1.2 |
| I6V | 325.15±3.5 | 2400 | 1.3 |
| I6V | 325.15±1.5 | 1600 | 1.5 |
| I8V | 448.00±7.2 | 3000 | 1.4 |
| I9V | 448.00±3.0 | 2000 | 1.6 |
| I10V | 448.00±1.4 | 1200 | 2.0 |
| I11V | 664.00±4.2 | 500 | 1.6 |

larization. The instrument will observe Earth from a mean altitude of 832 km with the sensor viewing angle 44.767° (measured from nadir). For all the channels, the mean footprint size is about 15 km, but the exact geo-location of samples differs. Therefore, a simultaneous utilization of data from different channels shall require remapping to a common footprint (Eriksson et al., 2020).

For this study, we conducted the forward simulations of the channels around: 183 GHz, 325 GHz, 448 GHz and 664 GHz (Table 2). For brevity, we assume that all simulations are mapped to a common footprint.

### 2.1.3   Small Microwave Satellite

The Small Microwave Satellite (SMS) is a hypothetical satellite which we introduce to represent the type of sensors currently being considered for future small missions carrying a single instrument. We assume it to be a single across-track scanning

microwave radiometer. In this study, we assume five 183 GHz channels and four 325 GHz channels, and just ignore if the mission has additional channels at lower frequencies or not. A brief summary of the channel specifications assumed is provided in Table 3.

### 2.2   Simulations

MWHS-2 simulated radiances during the period June–July 2020 are sourced from ECMWF. In the current version of the

ECMWF Integrated Forecasting System (IFS), cycle 47R1 (IFS, 2020), clear-sky and all-sky radiative transfer are performed simultaneously for monitoring purposes, despite all humidity sounders being assimilated via all-sky exclusively. These side by side radiative transfer calculations on a large variety of model scenes provides an ideal dataset for comparing radiances with





**Table 3.** Specifications of SMS channels.

| Channel | Frequency [GHz] | Bandwidth [MHz] | NEΔT [K] |
|---------|-----------------|-----------------|----------|
| SMS-1 | 176.31 | 2000 | 0.45 |
| SMS-2 | 178.81 | 2000 | 0.45 |
| SMS-3 | 180.31 | 1000 | 0.64 |
| SMS-4 | 181.51 | 1000 | 0.64 |
| SMS-5 | 182.31 | 500 | 0.88 |
| SMS-6 | 325.15±6.60 | 2800 | 0.60 |
| SMS-7 | 325.15±4.10 | 1800 | 0.75 |
| SMS-8 | 325.15±2.40 | 1200 | 0.92 |
| SMS-9 | 325.15±1.20 | 800 | 1.12 |

and without cloud effects. Out of all the available observations during the period, we use data for the latitudinal range: $60^\circ$ S to $60^\circ$ N and satellite zenith angle, less than $7.5^\circ$. With this filter, we have approximately 290 000 cases.

ICI and SMS frequencies are simulated with Atmospheric Radiative Transfer Simulator (ARTS, Buehler et al., 2018). For the forward simulations, Cloud Satellite (Cloudsat, Stephens et al., 2002) profiles during August 2015 are randomly selected. The input data are restricted between $60^\circ$S to $60^\circ$N, and surface is below 500 m. Both clear-sky and all-sky scenarios are simulated. The complete simulation setup is described in Appendix A. For ICI and SMS, 220 000 and 143 000 cases are simulated, respectively. For SMS, sensor viewing angles from $0^\circ$S to $45^\circ$N are simulated, but the results described in this

study are based on nadir viewing angle. Simulations for all three sensors are noise free, so to incorporate the measurement uncertainties, whenever needed, Gaussian noise is added according to the channel NEΔT (Table 1 – Table 3).

The simulations are split into training and testing datasets. The training dataset is used to train the machine learning model, while the testing dataset is used to evaluate the trained model. The construction and details of the model are described in Sect. 2.3. For MWHS-2, 220 000 simulations are randomly selected as training dataset, while 70 000 are used for testing. For

ICI, 175 000 cases are randomly picked to form the training set. The remaining 45 000 are used for testing. A smaller database is selected for SMS. 120 000 simulations are used for training and the remaining 23 000 for testing.

## 2.3   Quantile regression neural networks

The task that we aim to solve in this study is to predict the NFCS brightness temperature $y_{\mathrm{NFCS}}$ at a given 183 GHz channel from a vector of all-sky observations $\boldsymbol{y}$. Since the information-content of the cloud-contaminated observations is certainly too low

to solve this problem exactly, a probabilistic formulation is appropriate here. The aim thus becomes to predict the conditional distribution $p(y_{\mathrm{NFCS}}|\boldsymbol{y})$ of the NFCS brightness temperatures $y_{\mathrm{NFCS}}$ given the cloud-contaminated observations $\boldsymbol{y}$.





As has been shown in Pfreundschuh et al. (2018), QRNNs can be used to solve these type of problems. Instead of a point prediction, the QRNN is trained to predict a vector of quantiles of the distribution of the target variable conditional on the network input. Using these predicted quantiles, the cumulative distribution function (CDF) of the target variable can be estimated. QRNNs thus not only allow to predict a value $y_{\text{NFCS}}$ for the corrected brightness temperatures but also to estimate the uncertainty of the correction.

For this application, the predicted percentiles are chosen to be $0.2\,\%, 3\,\%, 16\,\%, 50\,\%, 85\,\%, 97\,\%$ and $99.8\,\%$. For a Gaussian distribution with mean $\mu$ and standard deviation $\sigma$, these quantiles approximately correspond to $\mu - 3\sigma, \mu - 2\sigma, \mu - \sigma, \mu, \mu + \sigma, \mu + 2\sigma, \mu + 3\sigma$ and thus allows estimation of the $\pm 1, \pm 2$ and $\pm 3\sigma$ confidence intervals.

QRNN's are trained to minimize the mean of the sum of the quantile loss functions,

$$\mathcal{L}_\tau(y_\tau, y) = \begin{cases} \tau |y - y_\tau|, & y_\tau < y \\ (1-\tau)|y_\tau - y|, & \text{otherwise} \end{cases} \tag{1}$$

for all selected quantile fractions $\tau$, where $y_\tau$ is the predicted quantile and $y$ the reference value from the training or test data. The quantile loss is also used in this study as a performance criterion for the tuning of the hyper-parameters of the QRNN (see Appendix B). In addition to the quantile loss, also the Continuously Ranked Probability Score (CRPS) is considered. Given a predicted CDF $F$ and the reference value $y$, the CRPS is defined as

$$\text{CRPS}(F, y) = \int_{-\infty}^{\infty} (F(y') - y)^2 \, dy'. \tag{2}$$

To compute the CRPS for a prediction from a QRNN, the predicted quantiles are used to derive a piece-wise linear approximation of the CDF of the predicted distribution. Note that CRPS is only used to evaluate hyper-parameter tuning.

The implementation of QRNN is similar to the one described in Pfreundschuh et al. (2018), except that this version uses PyTorch (Paszke et al., 2017) instead of Keras (Chollet et al., 2015) to implement the underlying neural network. The implementation is available as a part of version of the Typhon software package (Lemke et al., 2020). The major challenge for implementing QRNN for the current application was to select high performing neural network architecture. This was obtained through grid search over different hyper-parameter configurations. The details are described in Appendix B.

### 2.3.1 QRNN model configurations

In the study, two QRNN configurations are formulated for cloud-correction. The basic construction of both is that a separate network is trained for each 183 GHz channel to correct, using certain input data. The input data is all-sky brightness temperatures from selected input channels and/or additional data like land/sea mask. The output in both configurations is the posterior distribution of $y_{\text{NFCS}}$ for the target 183 GHz channel. The two configurations differ only by the number of input 183 GHz channels used in the training process:

1. QRNN-single: In this configuration, the training input comprises of all-sky brightness temperatures from the target 183 GHz channel and other channels. Additional data is included, if relevant. No other 183 GHz channel is included.





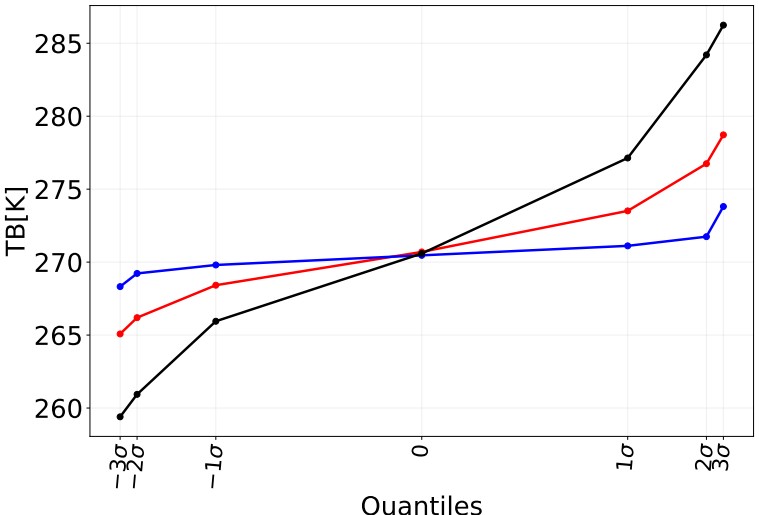

**Figure 1.** Examples showing the predicted quantiles of the conditional distribution of $y_{\text{NFCS}}$

2. QRNN-all: Same as QRNN-single, but all available 183 GHz channels are included.

For all three sensors described in this study, one or both of the above QRNN configurations are used. The selection of input channels is sensor dependent, and is described in detail when introduced later.

## 2.4 Evaluation metrics

QRNN predictions are posterior probability distribution of $y_{\text{NFCS}}$ described over the chosen quantiles. In order to facilitate the interpretation of results, examples of QRNN outputs are shown in Fig. 1. These examples illustrate the predicted quantiles for three different cases. The quantiles provide a quantification of the prediction uncertainty through a probabilistic upper and lower bound for each case. This is in contrast with other conventional correction/filtering methods, which give out only point estimates. However, for most applications only a single point estimate is required. In Bayesian analysis, usually the posterior mean or posterior median are selected as point estimates. In this study, we chose the posterior median as the best estimate for $y_{\text{NFCS}}$. To analyse the ability of QRNN in correctly predicting the point estimate, deviation of the median value from the corresponding true value ($y_{\text{NFCS}}^{\circ}$) is evaluated using common performance indicators like bias, mean absolute error (MAE) and standard deviation (STD). The asymmetry of error distributions around their mean is also calculated through the measure of skewness. For a univariate dataset of length $N$, the Fisher-Pearson coefficient of skewness is defined as:

$$g_1 = \frac{\sqrt{N(N-1)}}{N-2} \frac{\sum_{i=1}^{N}(Y_i - \bar{Y})^3/N}{\sigma^3}, \tag{3}$$

where $\bar{Y}$ and $\sigma$ are mean and standard deviation of the deviations, respectively.

For probabilistic predictions, accuracy of the point estimate is inadequate to gauge the complete performance. In a successful QRNN training, QRNN learns to predict not only an accurate point estimate but also the correct underlying uncertainty. An





ideal QRNN output should be sharp or in other words, all predicted quantiles should be concentrated in the vicinity of the point estimate. Nevertheless, the predicted posterior distribution should also be well calibrated, that is, the predicted distribution should reflect actually observed frequencies. A straightforward way to compare the two distributions is to plot the frequency of predictions and frequency of the true value in different prediction intervals. This is also commonly known as calibration plot. In a well-calibrated QRNN model, the calibration plot should follow the straight line $y = x$. Another way to assess how well

the predicted posterior distribution reflects the observed errors is to compare the predicted and observed errors. The predicted error is the deviation of a random sample drawn from the posterior distribution to its median. In this study, we analyse both the calibration plot and the predicted errors to assess the correctness of predicted uncertainties.

All evaluation results except hyper-parameter tuning, described in the study, have been made on the test dataset. The hyper-parameter tuning is made on the validation dataset (see Appendix B). The validation dataset is a separate part of the training

dataset which is held back during the training.

## 3 Correcting cloud affected data in MWHS-2

In this section, we introduce the QRNN based cloud correction in the context of current operational sensors. We use MWHS-2 to demonstrate the results. The choice is motivated by the fact that MWHS-2 has five complementary 183 GHz channels, along with additional 118 GHz channels. In order to formulate and test the correction approach, multiple QRNN experiments are

210 performed for MWHS-2. However, for brevity we show the comparison of the comprehensive results only for channel 14. Later the optimal experiment is extended to other 183 GHz channels. A brief comparison of the results is also made against existing cloud filtering methods. Further, the estimates of case-specific uncertainties obtained from QRNN are also evaluated.

### 3.1 Experiments

For MWHS-2, multiple experiments using both QRNN configurations are performed. With these we aim to delve into the

215 sensitivity of the method to different input channels:

1. In the first experiment, we examine the performance of QRNN cloud correction with MWHS-2 window channels: 89 and 150 GHz. In ECMWF NWP system, the differences between the observations of these two window channels are used to identify the cloud affected data for humidity sounding channels (Geer et al., 2014). To investigate the potential impact of these two window channels in QRNN based cloud correction, the configuration QRNN-single is applied. The

220 training inputs include all-sky brightness temperatures from target 183 GHz channel, 89 GHz and 150 GHz. Both 89 and 150 GHz are window channels so the land sea mask is also included as a training input. For example, for channel 14, the training inputs are all-sky brightness temperatures from channels 14, 1, 10 and land/sea mask. This combination is referred as 89+150 GHz in the text.

2. In the second experiment, we explore if few of low peaking channels of 118 GHz could have any potential in

cloud correction. To explore their impact, QRNN-single is trained with data from target 183 GHz, 89 GHz, 150 GHz,





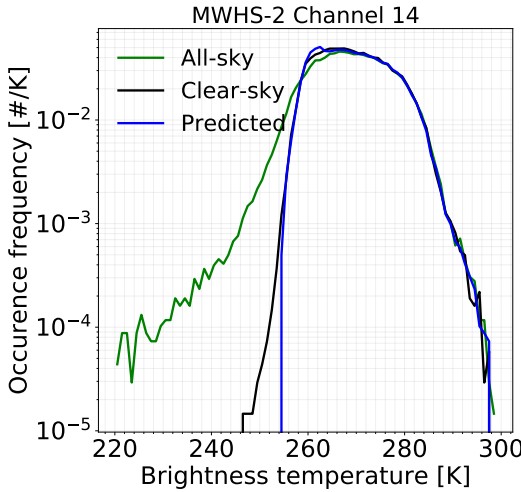

**Figure 2.** The distribution of point estimates ("Predicted") obtained from QRNN-single 89+150 GHz for MWHS-2 channel 14. The corresponding distributions for all-sky and clear-sky simulations are also shown.

118.75±1.1 GHz (channel 6), 118.75±2.5 GHz (channel 7) and land/sea mask. This combination is denoted as 89+150+118 GHz.

3. The third experiment is designed to assess the exclusive impact of 150 GHz in cloud correction. This experiment is motivated by the fact that hydrometeor impact at 150 GHz is strongest as compared to 89 GHz and other 118 GHz channels; and is less affected by surface emissivity. In this experiment, QRNN-single is trained with brightness temperatures from 150 GHz along with the target channel and land/sea mask.

4. The fourth experiment is based on the configuration QRNN-all. In this experiment, we use 89 and 150 GHz channels along with all 183 GHz channels to train QRNN. The use of 183 GHz channels for "self" cloud filtering has been studied by Buehler et al. (2007). They show that brightness temperatures between outer and inner humidity channels can be used as a criterion for cloud filtering. With QRNN-all, we investigate if additional humidity channels in the training process can improve the performance. Note that though the training inputs are same for each 183 GHz channel, the output is the target 183 GHz channel; thus each channel still needs to be trained separately. Land/sea mask is also included in the training. This combination is denoted as 89+150+183 GHz.

## 3.2 Prediction accuracy

### 3.2.1 QRNN-single applied to MWHS-2 channel 14

Posterior distributions of $y_{NFCS}$ obtained from experiment 89+150 GHz are similar to the ones shown in Fig. 1 and the distribution of point estimates is displayed in Fig. 2. The predicted values are able to correct most of the low brightness temperature cases, and overall a good match with the NFCS simulations is observed. However, QRNN is unable to predict the lowest





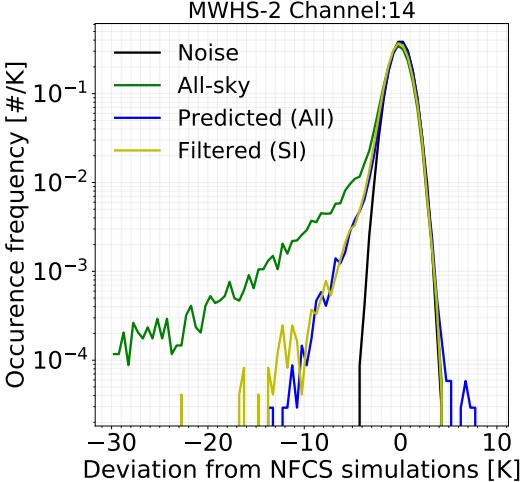

**Figure 3.** The error distribution for deviations of point estimates from NFCS simulations. The results are from QRNN-single experiment 89+150 GHz and MWHS-2 channel 14. Noise is also plotted for reference. The label "All-sky" represents the all-sky simulations, "Predicted (All)" denotes the predicted point estimates. The error distribution achieved with scattering index (SI) filtering is also shown (Filtered (SI)).

**Table 4.** The error statistics for deviations of point estimates from NFCS simulations. Results are for different QRNN experiments for MWHS-2 channel 14 (see Sect. 3.1). The statistics for all-sky and clear-sky simulations are also provided. The label "All" denotes the entire dataset of predicted point estimates, while "Pred. (5 K)" refers to the predicted point estimates but where cases with cloud correction greater than 5 K are excluded. The last two columns show the statistics obtained after filtering cloudy cases according to scattering index (SI) and scheme by Buehler et al. (2007) (B183). Bias, MAE, STD are in K, skewness is dimensionless.

| | Simulations | | QRNN-single | | | | | | QRNN-all | | Pure filtering | |
|---|---|---|---|---|---|---|---|---|---|---|---|---|
| | Clear-sky | All-sky | 89+150 GHz | | 89+150+118 GHz | | 150 GHz | | 89+150+183 GHz | | SI | B183 |
| | | | All | Pred. (5 K) | All | Pred. (5 K) | All | Pred. (5 K) | All | Pred. (5 K) | | |
| Bias | 0.00 | −0.84 | −0.10 | −0.09 | −0.05 | −0.04 | −0.11 | −0.10 | −0.10 | −0.09 | 0.24 | −0.52 |
| MAE | 0.80 | 1.45 | 0.89 | 0.87 | 0.90 | 0.88 | 0.90 | 0.88 | 0.62 | 0.60 | 0.92 | 1.15 |
| STD | 1.00 | 3.73 | 1.20 | 1.16 | 1.21 | 1.16 | 1.22 | 1.18 | 0.91 | 0.85 | 1.26 | 1.86 |
| Skewness | −0.02 | −12.72 | −1.17 | −1.00 | −1.06 | −0.90 | −1.16 | −1.04 | −1.78 | −1.71 | -1.95 | −3.45 |
| Rejection | - | - | - | 3.3 % | - | 3.4 % | - | 3.2 % | - | 3.3 % | 28.8 % | 3.5 % |

clear-sky brightness temperatures, and cases with brightness temperatures around 260 K, occur too frequently. The deviations
of point estimates from NFCS simulations are shown in Fig. 3. The large negative deviations are removed (blue curve), but residual cloud impact is evident in the negative tail. Most of these residual cases have departures less than 10 K. The appearance of a small positive tail also indicates overestimation in few cases. The corresponding error statistics (see Sect. 2.4) are provided in Table 4. In the uncorrected all-sky simulations, the negative departures due to cloud impact lead to a large negative skewness





(−12.72) and high bias (−0.84 K). QRNN trained with 89 and 150 GHz successfully corrects a major portion of the cloud
affected cases, and the bias is reduced to −0.10 K. However, negative skewness indicates presence of uncorrected departures.
Including 118 GHz in the training gives a small improvement compared to 89+150 GHz alone. The error distribution has lower
bias and is more symmetric, but the MAE and standard deviation remain unaltered. This indicates that the information from
118 GHz channels can be beneficial in predicting few cases correctly, but the overall performance is not exceptionally different
from 89+150 GHz. Similar is the case with using only 150 GHz—the differences between the three experiments are negligible.

Further, we investigate whether filtering the predictions with low accuracy could help in improving the error distributions.
For filtering such cases, we assume that predictions with cloud correction greater than 5 K are associated with large deviations.
In all three experiments, removing the cases with correction greater than 5 K removes around 3 % of the data, but only a
marginal positive impact on the accuracy is observed. The persistent negative skewness, even after filtering, indicates presence
of cases with residual cloud impact. Such cases are most likely to be associated with low or medium cloud impact. Choosing
a lower threshold can help in removing more partially corrected cases but at the cost of rejecting clear-sky cases. Since QRNN
gives out NFCS values, choosing an unusually low correction threshold can also classify noisy clear-sky cases as cloudy. For
example, for 89+150 GHz, a threshold of 1.5 K rejects almost 10 % of data, but the negative skewness is still not completely
removed.

In spite the fact that QRNN only provides a partial cloud correction, the results for MWHS-2 channel 14 are better than
265 what we achieve with existing cloud filtering techniques like scattering index (SI) and the filtering scheme by Buehler et al.
(2007), hereafter B183. SI uses the differences of brightness temperatures between 89 and 150 GHz to identify cloud affected
data; whilst in B183, they recommend a viewing angle dependent brightness temperature threshold at 183.31 ± 1.00 GHz and
brightness temperature difference between 183.31±3.00 GHz and 183.31±1.00 GHz as a measure of cloud impact. The results
obtained with SI and B183 are displayed in last two columns of Table 4. With SI threshold 5 K, more than 28 % of the data is
270 rejected, yet the resulting error distributions are poorer as compared to QRNN. The low bias and skewness values indicate that
most of the high negative departures are removed, but cases with low cloud impact pass the filter as clear. Similar is the case
with B183. Here only 3 % of the data is filtered out but the overall statistics are worse than both QRNN and SI. The results
from the two filters are not surprising as both are partial filters and aim at removing only cases with high ice content. The low
hydrometeor impact cases remain unaltered.

The three experiments were also performed for the other four 183 GHz channels, and a similar performance was obtained
(not shown). The positive effect of 118 GHz was slightly higher for MWHS-2 channel 15, but for others, no notable effect
was observed. In view of negligible performance differences between the three experiments, we consider the combination
89+150 GHz to be optimal. The results for other channels with this experiment are provided in Sect. 3.2.3.

### 3.2.2 Comparison of QRNN-all and QRNN-single

To assess the differences between the capabilities of QRNN-all against QRNN-single, we compare the error statistics obtained
for 89+150 GHz and 89+150+183 GHz for channel 14. Table 4 also shows the error statistics for the experiment QRNN-all. In
comparison to QRNN-single, we obtain almost similar error bias with QRNN-all, but the MAE and standard deviation reduce


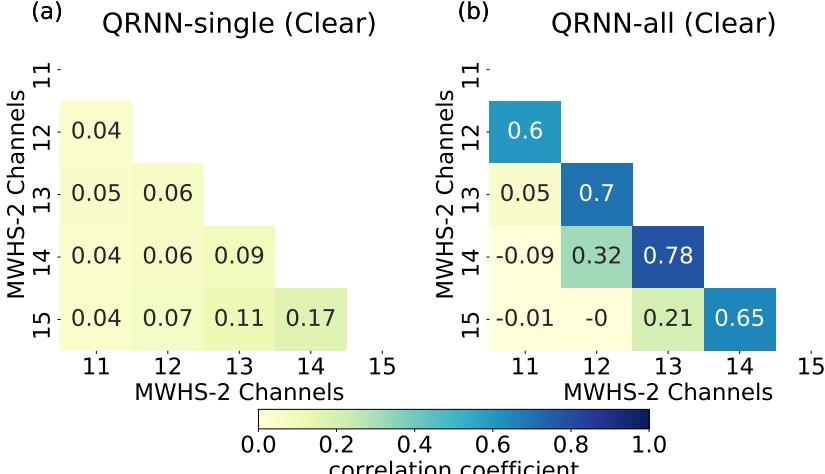

**Figure 4.** The triangular error correlation matrix obtained from (a) QRNN-single and (b) QRNN-all for MWHS-2. The label "Clear" represents cases with cloud impact less than 2 K.

by almost 30 % and 24 % respectively. However, the negative tail becomes more prominent. The low standard deviation, but strong negative tail indicates that the narrow spread is a consequence of correction of noise in clear-sky cases. Since a majority

of the cases are clear-sky, their impact dominates the whole statistics. In order to probe the positive effect of QRNN-all, we also estimate the errors for all cases with cloud correction greater than 5 K (table not shown). For such cases, QRNN-all has slightly better accuracy than QRNN-single. The bias in QRNN-all and standard deviation is $-0.53$ K and $1.93$ K, respectively, in comparison to $-0.61$ K and $2.04$ K as observed in QRNN-single. Thus, the concurrent use of all 183 GHz channels can provide additional information on cloud structures to QRNN.

Even though QRNN-all gives slightly better prediction accuracy, its inherent construction makes it crucial to examine the correlation between observed errors. Figure 4 illustrates the correlation matrix for both QRNN-all and QRNN-single. For clear cases, the observed errors (noise) in QRNN-single are uncorrelated between the five channels. However QRNN-all gives out highly correlated errors. The correlations are highest between adjacent channels, and drop out as the spacing between the channels increases. For cloudy cases, the observed errors obtained from QRNN-single are slightly correlated, but with

QRNN-all a very strong correlation is observed (not shown).

### 3.2.3   QRNN-single applied to channel 11, 12, 13 and 15

In this section, we extend QRNN-single to predict $y_{\mathrm{NFCS}}$ for MWHS-2 channels 11, 12, 13, and 15. The experiment QRNN-single with combination 89+150 GHz is used and the results are displayed in Table 5.

For channel 11, the bias after correction is $-0.12$ K in comparison to $0.15$ K in the all-sky simulations. The decrease in

bias is not significantly high, but the strong negative tail diminishes after correction, indicating removal of cases with large deviations. Nonetheless, the non-zero negative skewness also indicates presence of cases with residual cloud impact in the





**Table 5.** As Table 4, but for MWHS-2 channels 11, 12, 13, 15, and experiment QRNN-single 89+150 GHz.

| | | Simulations | | QRNN-single | | Pure filtering | |
| --- | --- | --- | --- | --- | --- | --- | --- |
| | | Clear-sky | All-sky | 89+150 GHz | | SI | B183 |
| | | | | All | Pred. (5 K) | | |
| Channel 11 | Bias | −0.00 | −0.15 | −0.12 | −0.12 | −0.04 | −0.05 |
| | MAE | 0.80 | 0.88 | 0.80 | 0.80 | 0.81 | 0.81 |
| | STD | 1.00 | 1.51 | 1.03 | 1.02 | 1.02 | 1.03 |
| | Skewness | 0.01 | −15.73 | −0.61 | −0.45 | −0.07 | −0.29 |
| | Rejection | - | - | - | 0.2 % | 28.8 % | 3.5 % |
| Channel 12 | Bias | −0.01 | −0.29 | −0.12 | −0.11 | −0.08 | −0.16 |
| | MAE | 0.80 | 0.98 | 0.82 | 0.81 | 0.82 | 0.87 |
| | STD | 1.00 | 2.02 | 1.08 | 1.06 | 1.04 | 1.15 |
| | Skewness | −0.01 | −17.27 | −0.96 | −0.79 | −0.25 | −1.02 |
| | Rejection | - | - | - | 0.6 % | 28.8 % | 3.5 % |
| Channel 13 | Bias | 0.00 | −0.53 | −0.11 | −0.10 | −0.14 | −0.31 |
| | MAE | 0.80 | 1.18 | 0.85 | 0.84 | 0.86 | 0.98 |
| | STD | 1.00 | 2.83 | 1.15 | 1.12 | 1.12 | 1.41 |
| | Skewness | 0.01 | −15.24 | −1.22 | −1.08 | −1.03 | −2.30 |
| | Rejection | - | - | - | 1.6 % | 28.8 % | 3.5 % |
| Channel 15 | Bias | 0.00 | −1.28 | −0.09 | −0.07 | −0.33 | −0.83 |
| | MAE | 0.80 | 1.88 | 0.98 | 0.93 | 1.03 | 1.46 |
| | STD | 1.00 | 4.97 | 1.36 | 1.27 | 1.52 | 2.69 |
| | Skewness | 0.00 | −10.11 | −0.69 | −0.82 | −2.87 | −4.33 |
| | Rejection | - | - | - | 5.5 % | 28.8 % | 3.5 % |

corrected dataset. Filtering the cases with high cloud impact has only a marginal positive effect. A similar performance is evident for channel 12. Correction reduces the bias to −0.12 K from −0.29 K and standard deviation to 1.08 K from 2.02 K̇. The MAE is approximately 16 % lower after correction, but still the negative skewness is not removed completely. For channel 13 and 15 also, a similar performance is seen. But in the latter, error distributions are more symmetric and have the largest spread as compared to other four channels. The effect of poor predictions is highest in channel 15 owing to its maximum sensitivity to hydrometeor impact.

A comparison with SI based filtering and B183 is displayed in last columns of Table 5. For channel 11 the performance of QRNN is comparable to both SI and B183. Similar is the case with channel 12 and channel 13, though the results from B183 are slightly poorer. The higher peaking channels are mostly transparent to hydrometeor impact, and the filtering schemes work



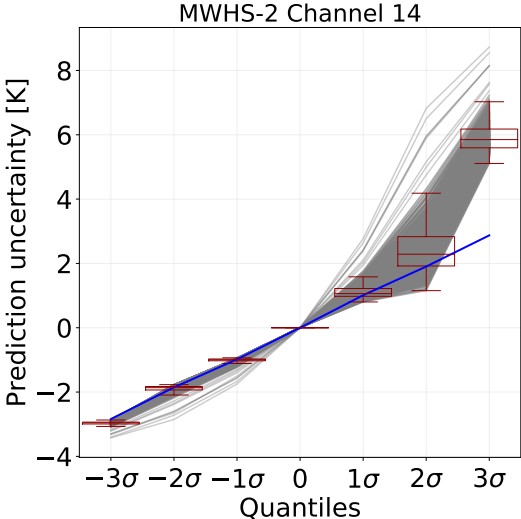

**Figure 5.** Examples of prediction uncertainties obtained from QRNN-single (89+150 GHz) for MWHS-2 channel 14. 1500 randomly selected cases are shown. The quantiles have been plotted at equidistant points. The blue line represents a Gaussian distribution with a standard deviation of 1.0 K. For each quantile, the sample variation is also shown as box plot.

well. The major caveat is the rejection of clear cases. With comparable accuracy, the fraction of rejection in SI is more than 28 % in comparison to only 3 % in B183. For channel, 15 the error statistics obtained with SI are slightly inferior in comparison to QRNN. The results with B183 are even worse. For all channels, the two filters succeed in removing the high ice cloud cases, but for lower peaking channels, the high negative skewness values indicate presence of cases with low cloud impact, which pass

the filter as "clear". Clearly the "one for all" approach of both filters is not adequate to cater for channel specific hydrometeor impact.

### 3.3 Prediction uncertainty

The quantiles given out by QRNN can be used to construct the probability distribution of the predictions in contrast to other correction approaches which give out only point estimates. Examples of the uncertainties given out by QRNN are shown in

Fig. 5. The spread of error distribution is asymmetric. The predictions over quantiles $-3\sigma$, $-2\sigma$ and $-1\sigma$ are quite sharp and lie close to the median value. In contrast, the spread of predictions over quantiles $1\sigma$, $2\sigma$ and $3\sigma$ is wider. The box plots indicate that cases with very high uncertainty occur infrequently. The highly uncertain predictions are mostly cloudy cases with low accuracy, but clear cases with high uncertainty could also be present. Also, all quantiles but $3\sigma$, follow the Gaussian distribution.

The calibration of the prediction intervals given out by QRNN is displayed in Fig. 6. We also analyse the calibration of the observational error model used for MWHS-2 in ECMWF NWP system (Lawrence et al., 2018). A gaussian error model is used to represent the distribution of the errors, and plotted under the label "SI". For the entire dataset, the predictions from QRNN





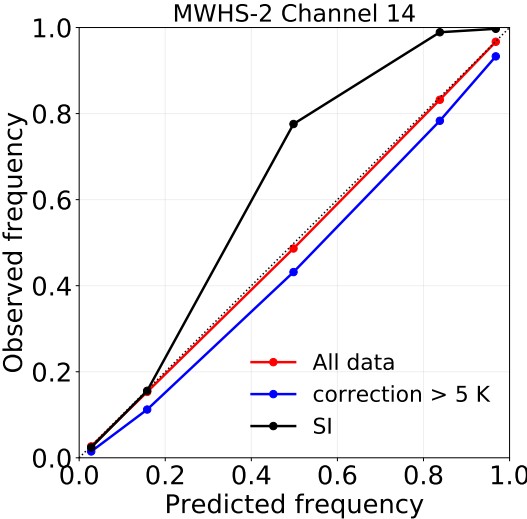

**Figure 6.** Calibration of the prediction intervals derived from QRNN-single (89+150 GHz) for MWHS-2 channel 14, and prediction intervals derived from Gaussian error model for ECMWF NWP system (SI). For the former, the calibration for both the entire predicted dataset and the subset with correction greater than 5 K are shown.

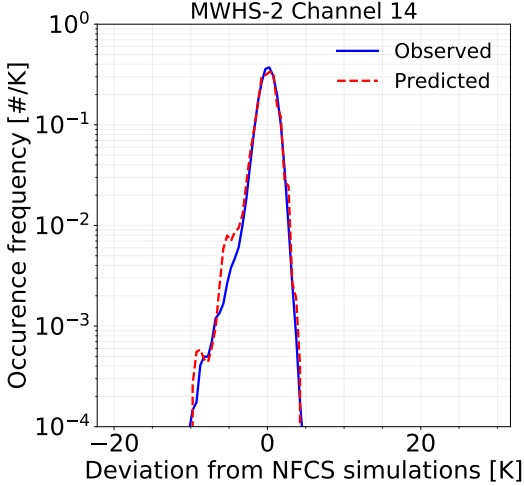

**Figure 7.** The distribution of predicted errors and observed errors obtained from QRNN-single (89+150 GHz) for MWHS-2 channel 14. The predicted errors are estimated as deviation of random samples from a posteriori distribution to corresponding median values.

follow the $y = x$ line, i.e. the predicted uncertainties are perfectly calibrated with the errors observed on the test data. However for the cases with correction greater than 5 K, the distribution is poorly calibrated, and the curve lies below the $y = x$ line, indicating that the prediction intervals are too narrow. This is in agreement with the wider spread of uncertainties for cases with low accuracy (Fig. 5). However, such cases form less than 6 % of the dataset. On the other hand, the calibration of ECMWF





error model is above the diagonal for predicted probabilities above 0.2. This suggests that the true probability is higher than what is predicted on these intervals.

Further, we analyse, if the predicted errors obtained from QRNN are representative of observed errors (Fig. 7). Both error

distributions are asymmetric, and this is in fact covered by the percentile distribution in Fig. 5. The predicted errors are slightly overestimated for negative values, but overall the predicted errors from the QRNN posterior distribution and the observed errors have a good match. This is in agreement with the perfect calibration seen in Fig. 6. It should be noted that the density plot is curtailed at $10^{-4}$. With a test dataset of 70 000 samples, we cannot represent the far wings of the distribution accurately. The high errors which we try to estimate are rare, and cannot be fully represented by QRNN. Note that since we do not derive any

sample from outside $\pm 3\sigma$, such cases could also belong to 0.003 % population not represented by the quantiles.

For other humidity channels, the predicted uncertainties followed similar behaviour and are not shown.

## 4 Correcting cloud affected data using sub-mm frequencies

In this section, we demonstrate that sub-mm channels can be used to formulate the cloud correction of data measured around 183 GHz. Results from different QRNN experiments with varying input conditions are described. The results are presented in

context of different sensors. Furthermore, the case specific uncertainties are also discussed.

### 4.1 Experiments

Two QRNN experiments are performed to investigate the efficacy of sub-mm channels in cloud correction:

1. In the first experiment, we apply QRNN-single configuration for cloud correction at three ICI humidity channels. In this case, the training data is the target 183 GHz channel and from all frequencies centered around 325 GHz, 448 GHz

and 664 GHz. For 664 GHz only vertical polarisation is included. No other data are considered. The experiment is also channel specific. For example, to predict NFCS values for channel I1V, the input training dataset includes noisy all-sky simulations from channels I1V, I5V, I6V, I7V, I8V, I9V, I10V, and I11V and the target is NFCS simulations for channel I1V.

2. In the second experiment, we investigate the possibility of using only channels around 325 GHz for cloud correction

at 183 GHz. This special case of utilizing only 325 GHz channels can be relevant for smaller satellite missions, as represented by SMS, where higher sub-mm channels are not available. In this experiment, QRNN-single configuration is used and it is trained with all-sky simulations from all 325 GHz frequencies from SMS and the target 183 GHz channel. For example, for the target SMS-1, the training inputs are SMS-1, SMS-6, SMS-7, SMS-8, SMS-9. QRNN is trained five times for each 183 GHz channel as target.




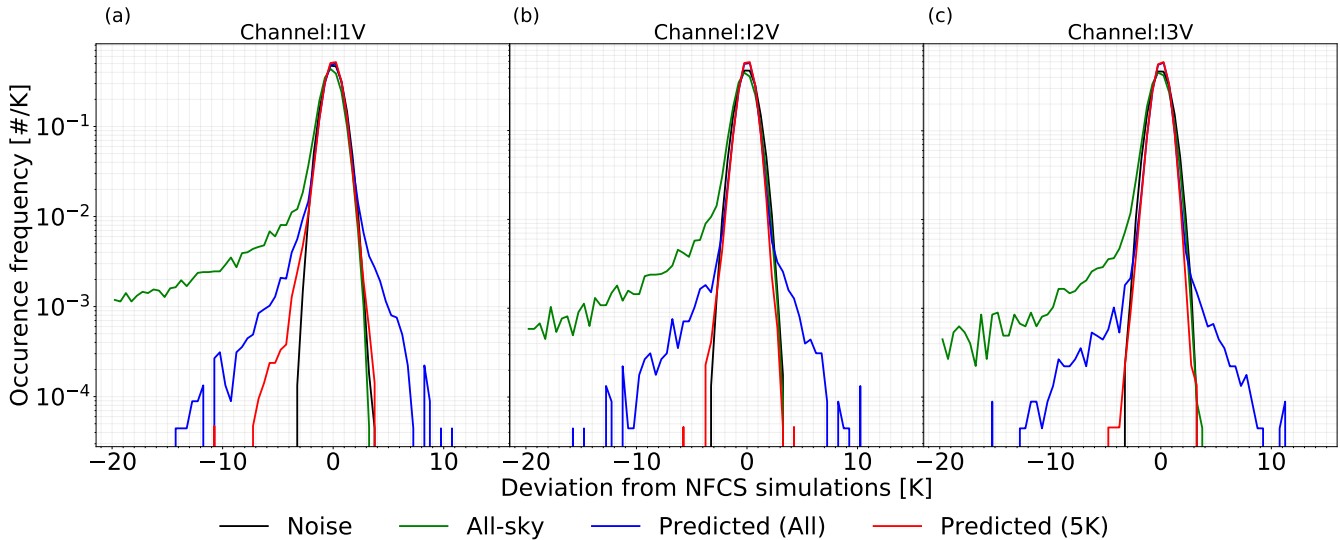

**Figure 8.** Same as Fig. 3, but from QRNN-single experiment for ICI channels (a) I1V, (b) I2V and (c) I3V.

**Table 6.** Same as Table 5, but from QRNN-single for ICI channels I1V, I2V and I3V. The fraction of rejected cases are given in parentheses.

| | | Simulations | | QRNN-single | |
| --- | --- | --- | --- | --- | --- |
| | | Clear-sky | All-sky | Pred. | |
| | | | | (All) | (5 K) |
| I1V | Bias | 0.00 | −1.87 | −0.02 | −0.00 (6.1 %) |
| | MAE | 0.64 | 2.32 | 0.70 | 0.60 |
| | STD | 0.80 | 8.84 | 1.06 | 0.79 |
| | Skewness | −0.01 | −8.10 | −1.51 | −0.64 |
| I2V | Bias | 0.00 | −1.04 | 0.00 | 0.01 (3.6 %) |
| | MAE | 0.64 | 1.53 | 0.57 | 0.51 |
| | STD | 0.80 | 5.95 | 0.86 | 0.65 |
| | Skewness | 0.00 | −10.79 | −1.85 | −0.22 |
| I3V | Bias | 0.01 | −0.63 | 0.02 | 0.02 (2.2 %) |
| | MAE | 0.64 | 1.15 | 0.54 | 0.50 |
| | STD | 0.80 | 4.27 | 0.80 | 0.63 |
| | Skewness | 0.01 | −13.37 | −1.51 | −0.13 |





### 4.2 Prediction accuracy

#### 4.2.1 ICI

The error distributions of the point estimates obtained from QRNN are shown in Fig. 8. The predicted values have symmetric error distributions albeit with a large spread. The large spread on the left is due to cases which end up with incomplete cloud correction, while the spread on the right is from cases where the predicted values are warmer than the simulations. For all three channels, quite similar behaviour is observed, though I1V has the most cases with residual cloud impact. If the predicted cases with correction more than 5 K are rejected, the resulting error distributions fit the measurement noise, except for I1V, where cases with residual cloud impact introduce a small negative bias. For a quantitative assessment of the errors, the results from various error metrics described in Sect. 2.4 are displayed in Table 6. The average bias in I1V all-sky simulations is $-1.87$ K, which reduces to $-0.02$ K after cloud correction. The corresponding standard deviation is 1.06 K in comparison to 8.84 K in the all-sky simulations. The prediction accuracy of QRNN is further higher, when filtering is made on the predictions. In this case, the residual bias is zero, and the standard deviation is 0.79 K, which is in fact of the order of measurement noise (0.80 K). Similar results are seen for I2V, though a better performance is observed. In I2V, the all-sky bias is $-1.04$ K which reduces to zero after correction and the MAE improves by almost 60 %. Removing cases with correction greater than 5 K from the predictions removes only 3.6 % of the data and reduces the absolute error further by 10 %. The standard deviation of the resulting dataset is only 0.65 K as compared to 0.80 K from noise. The reduction in the standard deviation is also evident in the Fig. 8, where the peak of distributions is sharper. In comparison to I1V and I2V, I3V has the lowest fraction of the cases with significant cloud impact. In the predicted dataset, the MAE is 0.54 K and standard deviation is 0.80 K. Filtering the cases with large correction reduces the MAE to 0.50 K and standard deviation to 0.63 K. For all three channels, the correction threshold filter successfully removes the cases with low accuracy. This is in contrast with results obtained with MWHS-2, where negative bias due to low cloudy cases is persistent even after filtering.

#### 4.2.2 SMS

In an analogy to the results from ICI channels, we perform a similar error distribution analysis and the results are displayed in Table 7. For channel SMS-1, the average bias and standard deviation in the uncorrected dataset is $-1.32$ K and 6.42 K respectively. However, after correction, the bias and standard deviation reduce to $-0.04$ K and 1.15 K, respectively. A decrease in the skewness of error distributions is evident, but a relatively high value after correction indicates presence of cases with partially-corrected cloud impact. Filtering out cases with 5 K correction improves the statistics but introduces asymmetry in the error distribution. This is most likely due to rejection of cases affected by over-estimation. For SMS-2, the predictions have slightly higher accuracy than SMS-1. The MAE in predictions is only 0.46 K in comparison to 1.24 K for the all-sky simulations. High skewness despite low bias (0.04 K) is most likely due to presence of isolated cases with large negative deviations. Nonetheless, filtering such cases makes the distribution more symmetric. For SMS-3 and SMS-4, again a similar behaviour is seen, though the distributions are more symmetric in the latter. For SMS-5, we obtain the most symmetric and narrow distributions after correction owing to the low sensitivity of SMS-5 to hydrometeor impact. Also, it is worth to noting





**Table 7.** Same as Table 5, but from QRNN-single for SMS channels. The fraction of rejected cases is given in parentheses.

| | | Simulations | | QRNN-single | |
| | | Clear-sky | All-sky | Pred. | |
| | | | | (All) | (5 K) |
|---|---|---|---|---|---|
| SMS-1 | Bias | 0.00 | −1.32 | −0.04 | −0.02 (4.71 %) |
| | MAE | 0.36 | 1.57 | 0.58 | 0.44 |
| | STD | 0.46 | 6.42 | 1.15 | 0.64 |
| | Skewness | 0.01 | −8.43 | −1.09 | −2.29 |
| SMS-2 | Bias | 0.00 | −1.00 | −0.04 | −0.02 (3.75 %) |
| | MAE | 0.36 | 1.24 | 0.46 | 0.37 |
| | STD | 0.45 | 5.17 | 0.87 | 0.52 |
| | Skewness | −0.01 | −9.59 | −3.25 | −1.26 |
| SMS-3 | Bias | 0.00 | −0.71 | −0.06 | −0.04 (2.86 %) |
| | MAE | 0.50 | 1.10 | 0.47 | 0.41 |
| | STD | 0.63 | 4.05 | 0.82 | 0.56 |
| | Skewness | −0.01 | −11.06 | −3.53 | −1.02 |
| SMS-4 | Bias | −0.00 | −0.45 | −0.03 | −0.03 (1.92%) |
| | MAE | 0.50 | 0.86 | 0.48 | 0.43 |
| | STD | 0.63 | 2.90 | 0.77 | 0.57 |
| | Skewness | 0.00 | −13.43 | −2.46 | −0.62 |
| SMS-5 | Bias | −0.01 | −0.29 | −0.04 | −0.04 (1.23 %) |
| | MAE | 0.71 | 0.90 | 0.63 | 0.60 |
| | STD | 0.89 | 2.17 | 0.88 | 0.77 |
| | Skewness | 0.00 | −13.54 | −1.32 | −0.27 |

that when cases with 5 K cloud correction are removed, the spread of prediction errors is narrower than noise for SMS-3, SMS-4 and SMS-5.

### 4.3  Prediction uncertainty

Similar to evaluation of uncertainty estimates for MWHS-2 (Sect. 3.3), we analyse the spread of predicted quantiles, their calibration and distribution of predicted errors.

Figure 9 shows the spread of prediction uncertainties over different quantiles for randomly chosen 1500 cases. The large spread in the predicted errors indicates that QRNN is successful in representing uncertainties for each case individually, rather





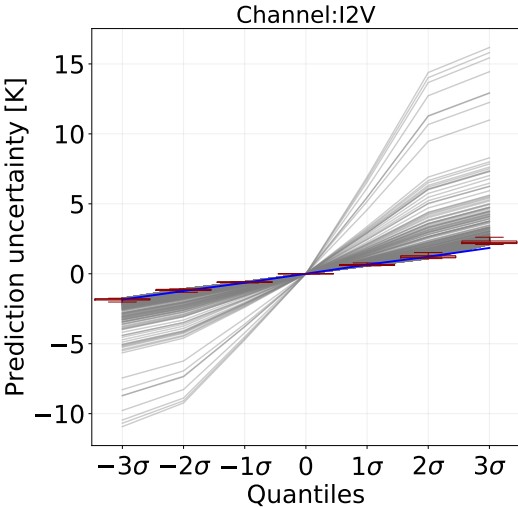

**Figure 9.** Same as Fig. 5, but from QRNN-single for ICI channel I2V. The blue line represents a Gaussian distribution with a standard deviation of $0.65\,\mathrm{K}$.

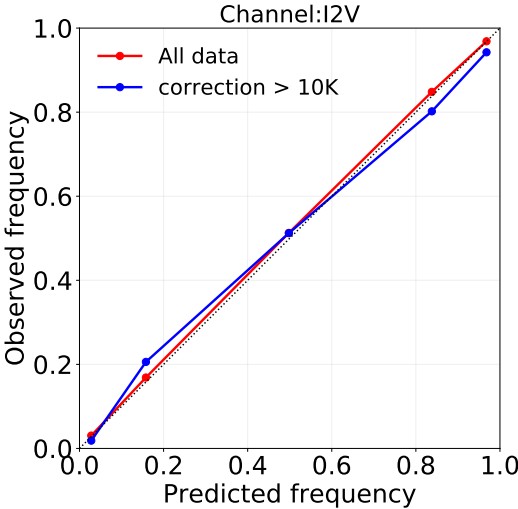

**Figure 10.** Same as Fig. 6, but from QRNN-single for ICI channel I2V.

than expressing them as a single measure. In the latter case, the uncertainty estimates would be concentrated along a narrow interval. Among the cases associated with low uncertainty, the distribution is quite symmetric along the median value. These cases are concentrated along a narrow interval and lie close to the blue line representing a Gaussian spread. On the contrast, cases with high uncertainty are unequally spaced and have a larger spread over positive quantiles than negative quantiles. The





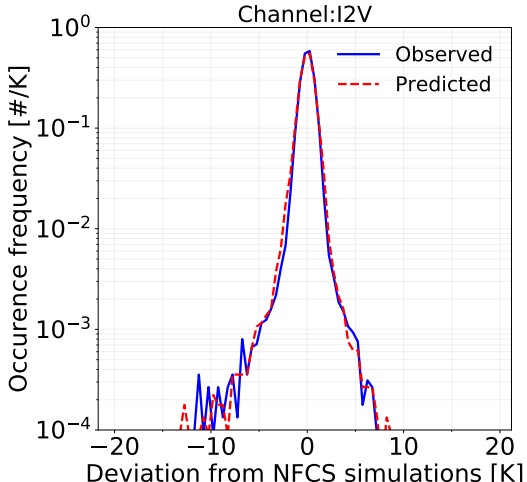

**Figure 11.** Same as Fig. 7, but from QRNN-single for ICI channel I2V.

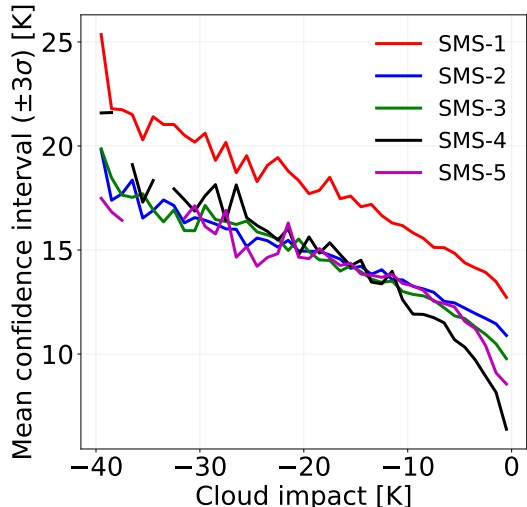

**Figure 12.** The average confidence intervals ($\pm 3\sigma$) plotted against the magnitude of cloud impact for all SMS channels.

narrow box plots also indicate that the majority of the predictions are sharp. These are clear-cases which dominate the dataset, while cloudy cases have more spread out uncertainties.

Figure 10 shows the calibration of the prediction intervals for I2V. The predictions for the complete test dataset are well calibrated and follow the $y = x$ curve. Similar is the case, when cases with correction greater than 5 K are considered (not shown). On the other hand, when the cases with cloud correction greater than 10 K are considered, the calibration is slightly worse. In spite of the fact that such cases are few (2 %), the high calibration indicates that QRNN is also successful in predicting





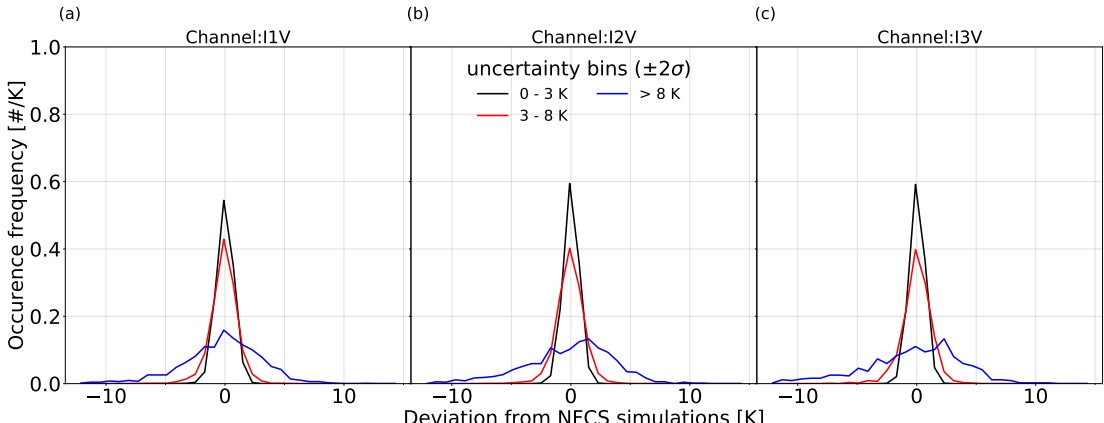

**Figure 13.** Distribution of errors binned according to their uncertainty. Results are from QRNN-single for channels I1V, I2V and I3V.

the uncertainties associated with rare cases. For other two channels also the predictions are well calibrated, except for cases with correction greater than 5 K in channel I1V (not shown).

Figure 11 shows the comparison of observed errors to predicted errors. The predicted and observed errors mostly have a good agreement but the predicted errors are spread out more asymmetrically towards the negative departures. QRNN is also not able to completely represent the wings of the distribution. This could also be a sampling issue, as the high errors we try to 415    predict constitute a very small part of the complete dataset.

With SMS, the behaviour of predicted uncertainties is observed to be similar as for ICI (not shown). However, the relation between the mean uncertainty estimate ($\pm 3\sigma$) and cloud impact is displayed in Fig. 12. For all five channels, the predictions with small or relatively low cloud signal have a low uncertainty or in other words have high sharpness. As fraction of cloud impact increases, the predictions become increasingly uncertain. The most uncertain predictions are for the lowest peaking 420    channel SMS-1, which incidentally is also most affected by hydrometeor impact.

To conclude the results, we analyse if the uncertainty estimates given by QRNN are representative of prediction accuracy. Figure 13 shows the observed ICI errors binned by their corresponding uncertainty in $\pm 2\sigma$ confidence interval. For all three channels, the spread of error distribution increases as the uncertainty about the accuracy of the prediction increases. The cases with high certainty have a narrow and sharp distribution, and the errors are mostly less than $\pm 2.5$ K. With increase in the 425    uncertainty, frequency of cases with high accuracy decreases and the distributions spread out symmetrically to higher errors. Poor predictions occur more frequently when uncertainty is high. Cases with accurate predictions yet high uncertainty are also present. In spite of individual variations in the error distributions for each channel, predictions and their corresponding uncertainties follow the same relationship. Similar results are also obtained with SMS (not shown).



## 5 Discussion

### 5.1 Cloud correction with existing sensors

The results from MWHS-2 show that QRNN based cloud correction is partially successful in correcting the cloud impact with existing humidity sounding sensors. The methodology can correctly address the large negative departures owing to cloud impact, but few cloudy cases end up with inadequate correction. The resulting error distributions are not completely symmetric but have a low bias and spread. Among several input channel combinations described for QRNN-single, the performance of combination 89+150 GHz is observed to be optimal. The positive performance with 150 GHz is not unexpected, as 150 GHz is sensitive to ice hydrometeors and cloud water; but using 89 GHz along with 150 GHz gives a slightly better performance. The channel 89 GHz is more affected by surface emission and is less sensitive to cloud water content, however its sensitivity to warm clouds in the lower troposphere could be important. We also investigate the impact of two temperature sounding channels (MWHS-2 channel 6 and 7). These channels provide complementary information to humidity channels in the lower troposphere. Including both these channels in the training process had no additional effect on the prediction accuracy and almost similar performance as with the combination 89+150 GHz is obtained.

Even though the cloud correction is partial, the performance is comparable or better to existing cloud filtering techniques like SI and B183. Note that both these techniques are a "one for all" approach for each 183 GHz channel; thus if one observation is classified as cloudy by the filter, it is removed in all humidity channels. This increases the probability of erroneously removing clear observations. For high peaking channels of MWHS-2, both SI and QRNN give almost similar results, but with almost 28 % rejection rate in the former. On the other hand, for low peaking channels, SI gives less accurate results than QRNN, as these channels have a stronger hydrometeor impact. This clearly indicates, that a channel specific approach like QRNN is more appropriate, and gives better performance.

The partial performance of QRNN is due to incomplete complementary information to 183 GHz channels. The weighting functions of window channels 89 and 150 GHz, and the two 118 GHz channels peak in the lower troposphere. Among these four channels, 150 GHz has the highest peaking function around 4 km (Chen and Bennartz, 2020). These channels can only provide coverage to the humidity channels in, lower and mid troposphere; nonetheless, the 183 GHz channels are sensitive to hydrometeor content up to 10 km. Due to missing complementary information from other channels in the upper troposphere, QRNN fails at predicting these cases accurately. Such cases are mostly associated with thin cirrus clouds, which have very small influence at 89 and 150 GHz. Without additional information from other channels we cannot expect QRNN to perform better. Among the other available channels, the overlapping weighting functions of 183 GHz can provide auxiliary information to train QRNN, but such information would be not be completely orthogonal or statistically uncorrelated. Results show that these channels indeed help in improving the training, yet non-orthogonality introduces highly correlated observational errors. The correlations for cloudy observations are not surprising as the cloud amount for different channels depends upon each other in a systematic way. However for clear-sky observations, the correlations should preferably be close to zero. This is observed to be true with QRNN-single, but QRNN-all fails at preserving the noise stochasticity. In the absence of hydrometeor impact, all 183 GHz channels provide same information to the learning model, introducing redundancy. Redundant patterns in machine





learning models often have undesirable effect on the predictive performance. In our application, redundant information does not affect the prediction accuracy, but introduces highly correlated observational errors between channels. Correlated errors are also undesirable for DA systems, but in future, if the operational centres progress with approaches dealing with correlated observation errors, for example as currently done for ATMS at ECMWF (Weston and Bormann, 2018), concurrent use of 89, 150 and 183 GHz channels would give the best cloud correction performance.

## 5.2 Cloud correction with sub-mm frequencies

In the ICI observations examined here, the results show that using sub-mm channels can successfully predict the NFCS values with very high accuracy for I2V, and I3V. The predicted values have an excellent match with the true values, and the departures are symmetrically distributed around zero mean. Few cases with high cloud impact do affect the accuracy and introduce a small negative bias, but such cases are easily filtered out with a simple correction threshold filter. For example, it is shown that filtering out cases with correction greater than 5 K, results in variabilities of the order of sensor noise with minimal reduction in data (2–6 %). For channel I1V, a slightly lower accuracy is observed due to relatively higher number of cases with residual cloud impact. The accuracy is improved by activating the correction filter, but some effect of residual cases is still apparent. Interestingly, reducing the correction threshold further has no significant effect on flagging these residual cases. In fact only clear-sky cases are removed. This is a consequence of the correction being too low. Since such cases introduce a small negative bias and skewness, they are more appropriately related to low-cloud impact. Compared to other two higher peaking channels, I1V is more sensitive to the effect of hydrometeors and contamination from surface effects (Fig. 4 of Eriksson et al., 2020). The cases with surface contamination are also localized and seasonal. The weighting functions of sub-mm channels can provide only a partial coverage to the hydrometeor impact at I1V. A part of lower troposphere sensed by I1V has almost zero coverage from sub-mm channels. Though such cases are few, their lack of representation prevents QRNN from correctly learning to predict the clear-sky values accurately.

A similar pattern is seen when only one 325 GHz channel is used to correct cloud impact in SMS. QRNN is successful in predicting NFCS values for all channels except for the lowest peaking channel SMS-1. For SMS-1, the resulting error distributions have significantly lower spread than the all-sky simulations, but are still negatively skewed. In spite of the slightly inferior performance for SMS-1, the high accuracy for other channels indicates that a single sub-mm channel like 325 GHz is also sufficient for cloud correction at 183 GHz. This is an important result as smaller satellites may be limited by their size to measure several sub-mm wavelengths.

With ICI and SMS, for some channels the variability of errors smaller than measurement noise is achieved. This is a consequence of predictions for cases which lack cloud impact. QRNN predictions are weighted mean of measurements between channels. In the absence of clouds, also the sub-mm channels provide humidity information that is incorporated in the 183 GHz NFCS estimate and some compensation of noise can be achieved. This effect is observed to be stronger in ICI than SMS, as the former has a higher number of channels giving redundant information. Note that with actual satellite measurements, the spread of error distributions smaller than sensor noise could be difficult to achieve due to other underlying uncertainties not considered here.





### 5.3 Prediction uncertainty and implications for data assimilation

Another advantage of QRNN is the estimation of case-specific uncertainties. That is, the predictions over chosen quantiles quantify the underlying uncertainty of the particular case, and not just represent some ensemble mean error. This has the consequence that a DA system can assign a proper weighting of each individual QRNN prediction. The analysis of QRNN predicted errors and calibration plots confirmed that QRNN is successful in providing well calibrated probabilistic predictions also in practice, except for few cases associated with high error. Poorly calibrated predictions are a consequence of outliers which are not well represented in the training dataset. The a priori distribution of the training dataset is dominated by clear-sky cases, however, the cloudy cases which occur infrequently or lack independent complementary information cannot be represented by same a priori distribution. The distribution of rare cases can be improved by increasing the training dataset size, but lack of complementary information can only be balanced by including additional training inputs, e.g. brightness temperatures for other channels.

The symmetric and low spread error distribution with uncertainty estimates is also an important result from the DA perspective. Most of the existing cloud filtering schemes work well only at removing cases with high cloud impact, and as a consequence, the error distributions are highly skewed. To use these observations correctly, DA schemes often inflate their assigned observation errors at the cost of artificially suppressing the observational impact. However, the symmetric error distributions obtained from QRNN allow effective utilization of almost complete data without the need for artificial error inflation. In fact for DA, filtering based on correction threshold would be needless as cases with low accuracy shall inevitably get down-weighted due to high uncertainty.

## 6 Conclusion and outlook

In this study, a methodology based on quantile regression neural network (QRNN) is used for identifying and correcting the cloud contamination in operational microwave humidity channels. QRNN is a neural network which trains on all-sky brightness temperatures from channels containing orthogonal information to humidity channels, to estimate the noise free clear-sky (NFCS) brightness temperatures. The output is the posterior distribution of predictions over different quantiles. QRNN is a channel specific approach, or in other words, QRNN is trained separately for each channel, and the cloud correction for each is independent of other channels.

The applicability of QRNN based correction to current sensors is demonstrated with MHWS-2 (MicroWave Humidity Sounder-2) and it is shown that QRNN is partially successful in removing the cloud impact. In comparison to existing clear-filtering approaches, QRNN gives comparable or better performance with minimal rejection of data. Nonetheless, since cloud correction using a limited number of microwave channels is an ill-posed problem, a point-estimate-based correction using only microwave observations between 89 and 150 GHz is inherently limited in its capability of correcting cloud-contaminated brightness temperatures.

Based on the promising results from MWHS-2, and with future scope, we extend the study to include data from Ice Cloud Imager (ICI) sub-millimetre (sub-mm) channels. The results show that with sub-mm channels, QRNN is able to correctly pre-





dict most of the cloudy cases and can provide high quality cloud corrected radiances. The predicted radiances have symmetric and narrow error distributions and for some channels the spread is smaller than the measurement noise. This makes it highly suitable for application to data assimilation (DA) systems. The robustness of sub-mm channels in cloud correction is also demonstrated with use of only 325 GHz for cloud correction. This is applicable to smaller satellite missions as represented by SMS, where only one of the higher frequencies is available. The results indicate that utilisation of only 325 GHz can also

be beneficial when other channels are absent. It's possible that the ICI sub-mm channels could also be used to cloud correct humidity radiances from MicroWave Imager (MWI)—another conically scanning radiometer onboard Metop-SG. MWI will measure frequencies from 18 GHz to 183 GHz. Both MWI and ICI have the same requirements for incidence angle and fore-view observations, but different footprints. Although re-mapping to a common footprint would slightly compromise the data quality, the high accuracy achieved with ICI simulations suggests that the QRNN would work well even when actual

measurements are available.

The biggest advantage of QRNN compared to other regression based approaches is its probabilistic nature. The QRNN predictions over chosen quantiles are a measure of the accuracy at different probability levels. In this study, the predicted quantiles given by QRNN work well in representing the accuracy of the point estimate. The point estimates with low error have high certainty and incorrect predictions have low confidence. In comparison to deterministic correction approaches,

the corrected radiances along with uncertainty estimates give additional benefit for DA systems. The statistical structures of underlying uncertainty are extremely important for DA systems as they offer a measure of reliability and robustness of observations.

The cloud corrected radiances have great potential in both retrieval schemes and numerical weather prediction (NWP) systems. Even with availability of new sensors and better observations, the problems posed by undetected cloud impact limit

the complete usage of humidity observations. However, with the cloud correction methodology presented here, we can aim at resolving these limitations. This is especially true for clear-sky assimilation systems, which reject up to 80 % of the available observations due to cloud contamination. In fact, one of reasons for the positive performance of all-sky assimilation systems is attributed to the larger number of assimilated observations in comparison to clear-sky observations. If QRNN can provide the clear-sky NWP systems with cloud cleared microwave radiances with minimal rejection of data, it may be possible to reap

forecast benefits without additional complexities and computational cost of scattering calculations. The all-sky assimilation systems could also benefit indirectly from cloud corrected radiances, to provide a measure of cloud impact as a diagnostic field for analysing increments. Another advantage of combining 183 GHz and sub-mm for cloud correction is that NWP systems, which are not yet prepared for higher frequency channels, could still benefit from the data early on.

In this study, we demonstrate the correction scheme with data from a limited period. The seasonal and latitudinal distributions

are not taken into account, and more complex surfaces such as sea ice, snow, and high orography have not been considered. It remains to be seen whether QRNN shows any seasonal sensitivity or dependency on cloud types. Such analysis could also be important at improving the a priori distribution of rare cases.





*Code availability.* QRNN is available as a part of the *typhon: tools for atmospheric research*, https://doi.org/10.5281/zenodo.3626449. The source code for all the analysis presented in the article is available as git repository (https://github.com/SEE-MOF/aws)

## Appendix A:  ARTS setup

All radiative transfer forward simulations are made by the Atmospheric Radiative Transfer System (ARTS, Eriksson et al., 2011; Buehler et al., 2018)), version arts-2.3.1. All simulations are based on dBZ-based model system presented by Ekelund et al. (2020), i.e. CloudSat reflectivities are used as input and are converted to ice water content (IWC) and rain water content (RWC) using microphysical assumptions. For each atmospheric case, both all-sky and clear-sky calculations are performed. In the former, all hydrometeors contents are set to zero, while in the latter, IWC and RWC derived from CloudSat reflectivities, and liquid water content (LWC) from ERA-Interim (Dee et al., 2011) are included. In order to avoid a possible bias between clear-sky and all-sky calculations for insignificant hydrometeor contents, both calculations are made by the RT4 solver (Evans and Stephens, 1995). The absorption model takes into account the effect from nitrogen (Rosenkranz, 1993), oxygen (Rosenkranz, 1993), liquid water content (LWC, Ellison, 2007), and water vapour. For water vapour, absorption model as in RTTOV (Radiative transfer for TOVS, Saunders et al., 2018) is followed, but with few modifications (Turner et al., 2019). LWC is assumed to be totally absorbing. In the mapping of CloudSat reflectivities to RWC and IWC, a total separation between liquid and ice phase is assumed. All scattering hydrometeors at temperatures above $0^{\circ}$ C and below $0^{\circ}$ C are assumed to be rain and ice hydrometeors respectively. For RWC, the particle size distribution (PSD) of Abel and Boutle (2012) is applied. The PSD of IWC follows the basic formulation applied in DARDAR (http://www.icare.univ-lille1.fr/projects/dardar Delanoë and Hogan, 2008), using latest parameter values (i.e. $\alpha$ and $\beta$) as given by Cazenave et al. (2019). This PSD can be considered as a "two moment" scheme, but is here applied in a one moment manner by setting $N_0^*$ (as a function of temperature) following Table 5 of Delanoë et al. (2014), and letting the radar reflectivity set the remaining moment. Single scattering data are taken from Eriksson et al. (2018). For ice hydrometeors, three habits are applied: perpendicular 3-bullet rosette, large plate aggregate and large column aggregate. In the last two cases, the aggregates are complemented with single crystal data to also cover smaller sizes. The particles are assumed to have a totally random orientation. To apply oriented particles is much more computationally costly and could not be accommodated inside the study. The land emissivity was taken from Tool to Estimate Land-Surface Emissivities at Microwave frequencies (TELSEM, Aires et al., 2011) and the Ocean/water from Tool to Estimate Sea-Surface Emissivity from Microwaves to sub-Millimeter waves (TESSEM, Prigent et al., 2017).

Using the setup described above, forward simulations are performed for ICI and SMS. The output from ARTS is first two elements of the Stokes vector, which are converted to brightness temperatures for H- and V-polarisation.

## Appendix B:  QRNN network structure

A high performing QRNN model also requires tuning of multiple hyper-parameters. These parameters determine the structure and the training set-up of the neural network. Several of these hyper-parameters are non-learnable/, and must be defined before





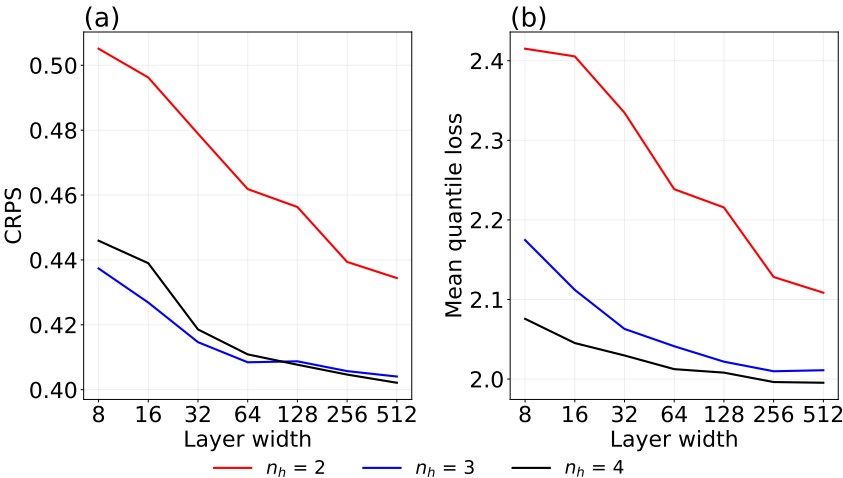

**Figure B1.** (a) CRPS and (b) mean quantile loss averaged over all predicted quantiles for different combinations of layer width and hidden layers ($n_h$). The results are from QRNN-single applied to I2V.

beginning of every training. Grid search is one of the most often employed techniques for hyper-parameter tuning. In grid
search, different combinations of hyper-parameters are selected and for each, the model performance is evaluated. The model
architecture with the best performance is selected. For the structural parameters, usually a grid search over the number of
neurons (width) and hidden layers (depth) is performed. The model is trained for multiple values of layer widths and hidden
layers, and the best configuration is selected by evaluating the predictions over validation data. Similarly, the training process
is optimized by performing a grid search different training parameters such as : batch size, learning rate, number of epochs etc.
We use quantile loss and CRPS for evaluation of the model performance.

In this study, we investigated the performance of QRNN only to certain hyper-parameters like number of neurons, hidden
layers, learning rate, convergence epochs and batch size. The optimization of other hyper-parameters was not performed and
were chosen empirically. Firstly, we performed a grid search to define the structure of the neural network. We evaluated the
performance for three sizes of hidden layers ($n_h$ = 2, 3, 4), and layer widths of sizes in the set [8, 16, 32, 64, 128, 256, 512].
The mean quantile loss and CRPS over all predicted quantiles was computed for each configuration (Fig. B1). Increasing
the complexity of the network by increasing the layer width and depth has a positive impact on performance. However for
four hidden layers, increasing the number of neurons beyond 128 has no significant impact on the performance. On basis of
these results, a neural network with four hidden layers and 128 neurons in each layer is selected. For optimising the training
parameters, a customised learning rate scheduler was implemented. The initial learning rate was reset after a certain number of
610 epochs. We started the training process with a initial learning rate of 0.1, and decreased it by a factor of 10 after 100 epochs.
The best neural network performance was obtained when the network was trained three times. Each time with a new initial
learning rate. For each training, if the validation loss remained unchanged till 6 training epochs, the learning rate was reduced
by a factor of 2. In order to select the batch size, we simply compared the performance for two batch sizes: 128 and 256, and





the former gave better results. Concerning number of epochs, we obtained best results when the network was trained longer.

Choosing a lower value of epochs (e.g. 50), did not affect the accuracy of the median value, yet deteriorated the prediction uncertainty. We did not optimise the type of activation function and Rectified Linear Unit (ReLu) was used in all layers.

Though these set of hyper-parameters were selected for QRNN-single applied to I2V, they worked well for both ICI and SMS QRNN experiments. However, for MWHS-2, an identical hyper-parameter framework but three hidden layers gave best results.

*Author contributions.* The study was conceptualised and supervised by PE. The analysis, validation and visualisation of results was done by IK. The QRNN package is made available by SP. ARTS bulk simulations were set up by SP, and MWHS-2 data was provided by DID. IK wrote the original draft, with contributions from PE, SP and DID. All authors contributed to the study with discussions and feedback.

*Competing interests.* The authors declare no conflict of interest.

*Acknowledgements.* The part associated with SMS was done with funding from EUMETSAT (contract EUM/CO/20/4600002417/CJA),
while remaining work was mainly funded by the Swedish National Space Agency (grants 166/18 and 154/19). David Duncan is supported by the EUMETSAT Fellowship Programme. This project would not had been possible without the contribution of all individuals behind the open source softwares used (ARTS, Python, PyTorch, Numpy and Matplotlib).





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
