# Peer review of "Can machine learning correct microwave humidity radiances for the influence of clouds?"

_Atmospheric Measurement Techniques, 2020_

## Referee Comment (RC1) · Anonymous Referee #1 · 11 Jan 2021

Kaur et al. manuscript, entitled "Can machine learning correct microwave humidity radiances for the influence of clouds?", investigates the use of machine learning toward the better detection of cloud-affected radiances from passive space-borne microwave radiometer. The manuscript is very well written and structured, interesting to read, and highly relevant to the climate community (from production to climate data record to reanalysis) and the NWP community (for the assimilation of microwave radiances). The authors demonstrate that a well-tuned quantile regression neural networks can detect and correct the impact of scattering from hydrometeor on microwave radiances with minimum data loss (compared to traditional cloud filter rejecting a large number of potentially useful information) while providing individual uncertainties. This is tested on simulations existing and future instruments.

[Figure]

I certainly recommend the publication of this work. Before that, I do have few minor comments that need clarification (see below) as well as some open questions whose response could be integrated to the manuscript or not, I leave that to the authors discretion.

Minor comments:

L28: "precipitation and most dense clouds" I think the hydrometeor size is an important factor.

L66: "only using measurements (no "background" data involved)" My understanding is that the method is in theory model-free, but for the demonstration in this study, simulations (e.g. background simulated at MWHS2 frequency) are used. Am I correct? Maybe this should be stressed here. Why not try with real MWHS2 observation for the all sky dataset?

L137 "Simulations for all three sensors are noise free, so to incorporate the measurement uncertainties, whenever needed, Gaussian noise is added according to the channel NEDT (Table 1 – Table 3)." Are the errors arising from the radiative transfer calculation accounted for?

L159: "for all selected quantile fractions " by quantile fraction, do you mean the n th amongst the 7 selected (from 0.2 to 99.8%)? Also, 16, 50 and 85% are not symmetric (rounding?)

L163: Pfreundschuh uses an indicator function I (=1 or 0) in the CRPS, the authors here use y, can they explain the difference?

L173: " The input data is all-sky brightness temperature" the simulated one, even for MWHS2, right?

L301: "0.15 K" should be -0.15

L453: "Among these four channels, 150 GHz has the highest peaking function around 4

km (Chen and Bennartz, 2020)" Could you please clarify what you mean here, 150GHz is neither the highest nor the lowest peaking channel nor it peaks at 4km. Channel 89, 118+/-1.1, 118+/-2.5, and 150 GHz peak at 0.1, 9.6, 2.9, 1km, respectively (according to Chen and Bennartz, 2020), this can also be seen in Lawrence et al. (2018) Fig. 1 through the Jacobians of channels 6 and 7.

L460: "but such information would be not be completely orthogonal" duplicate "be"

Table 1: NEDT are constructor specifications, the real noise is lower, see

Fig 5 Guo, Y., J. Y. He, S. Y. Gu, and N. M. Lu, 2019: Calibration and validation of Feng Yun-3-D microwave humidity sounder II. IEEE Geoscience and Remote Sensing Letters, doi:10.1109/LGRS.2019.2957403.

Tab 5 Carminati, F., Atkinson, N., Candy, B., Lu, Q.: Insights into the Microwave Instruments Onboard the Feng-Yun 3D Satellite: Data Quality and Assimilation in the Met Office NWP System. Adv. Atmos. Sci. (2020). https://doi.org/10.1007/s00376-020-0010-1

Questions:

Is this method applicable to IR e.g. to an ATOVS system?

The authors explain that it could benefit the all-sky assimilation systems indirectly for the analysis increment. Instead (or in addition) could it be used to model the variable observation error (when and by how much to be inflated)? This would be, in my view, the most valuable.

Could the uncertainty use for weak constraint 4dvar?

What is the resource cost of this method (is this fast enough to be used in 1-h regional nwp with 30min window)?

---

## Referee Comment (RC2) · Anonymous Referee #2 · 19 Jan 2021

Review of Kaur et al., "Can machine learning correct microwave humidity radiances for the influence of clouds?"

General Comments

The authors present a novel approach (quantile regression neural networks) to screening microwave brightness temperature observations near 183 GHz for cloud influence and constructing estimated "noise-free clear-sky" observations that could be used in data assimilation applications. The article is clear, concise, and well written, and the topic is relevant and important. I recommend publication after the following minor comments and corrections are addressed.

Minor Comments

[Figure]

As I understand it, this study uses only simulated measurements (either with or without hydrometeors included), even for MWHS-2 for which actual measurements are available. This should be made clearer, especially given line 66, which states that no "background" data is required for the method. I'd also like to see a bit more discussion about whether the results you present should be expected to hold for real data. An assumption that you are making is that your forward model can represent real clouds with enough fidelity that a model trained on simulated data will still work on real-world data. Can you point to any evidence to back up this assumption? Have you tried comparing the histograms of model-simulated Tbs with real-world MWHS-2 Tbs?

What is the computational cost of the QRNN method compared to simpler cloud-clearing filters? Could it feasibly be implemented into existing NWP models given current computational constraints?

Lines 135-143: Why the large difference in number of cases simulated for ICI (220,000) vs. AWS (143,000), if they are both coming from the same population of CloudSat profiles? For the training dataset, you use about 75% of total cases for MWHS-2, about 80% of cases for ICI, and about 84% of cases for AWS. Why this difference in proportions?

Line 340: "With a test dataset of 70 000 samples, we cannot represent the far wings of the distribution accurately." Couldn't one apply this same reasoning to Figures 2,3, or 8, which have density values below 10-4 included?

Figure 13: I wonder if there might be an easier, more concise way to evaluate whether the uncertainty intervals are properly calibrated. Namely, could you simply calculate how often the true value falls within the $\pm 2\sigma$ uncertainty range, for each of the uncertainty bins (0-3K, 3-8K, 8+K) that you've included on the plot? If this percentage is significantly less than 95%, it would suggests that your uncertainties are too small, while if it were closer to 100% it would suggest your uncertainties are too large. Even if you don't get rid of Fig. 13, I think this would be useful information to include in the

paper.

Lines 498-499: ". . . other underlying uncertainties not considered here." Whether here or elsewhere in the paper, I think you should talk a bit more about what these other uncertainties are (radiative transfer model errors should certainly be discussed), and what effects they might have on your results.

Typos

Line 26: The phrase "weather satellites are since some time equipped" is confusing to me. I suggest "weather satellites have for some time been equipped. . ."

Line 51: I believe you are missing the word "on" between "predicated" and "Gaussian."

Line 154: Should 85% be changed to 84%? That would be +2 sigma for a normal distribution.

Line 455: ". . . provide coverage to the humidity channels in, lower and mid troposphere" This is confusing – get rid of the comma and add the word "the" instead?

Line 494: "QRNN predictions are weighted mean. . ." I think this should say, "QRNN predictions are the weighted mean. . ."

Line 508: Missing "the" before "same"

---

## Short Comment (SC1) · 1 Feb 2021

With this note, we want to clarify that the original AMTD manuscript has been exchanged and why this has been done.

The change consists of that the small mission introduced in Sec 2.1.3 now is presented as a hypothetical (but representative) concept. The reason for this change is that the mission previously mentioned is in an evaluation and we were not supposed to possibly influence the selection procedure by presenting results associated with the mission. We consider the new manuscript version scientifically equal to the original version.

Patrick Eriksson and co-authors

---

## Author Comment (AC1) · 4 Mar 2021

**Response to Reviewer 1**

The authors would like to thank the reviewer for his/her careful review and constructive comments, which we believe will help improve the content of the manuscript. Below, under each bullet point, we provide a point by point response to each comment/question. The author responses are given in blue, and textual changes are italicised.

**General comments**

- "precipitation and most dense clouds" I think the hydrometeor size is an important factor.

    Reply: We agree that the hydrometeor size is an important aspect, but when it comes to an overview of the cloud contamination at $183\,\mathrm{GHz}$, precipitating clouds and clouds with large optical thickness have the strongest effect, and a dense cloud could be composed of hydrometeors of different shapes and sizes.

- L66: "only using measurements (no background data involved)" My understanding is that the method is in theory model-free, but for the demonstration in this study, simulations (e.g. background simulated at MWHS2 frequency) are used. Am I correct? Maybe this should be stressed here. Why not try with real MWHS2 observation for the all sky dataset?

    Reply: Yes, the demonstration of the entire concept is based on simulations. We do not try with the real MWHS-2 observations as the validation of the training process could only be performed against clear-sy simulations. To stress on the fact that only background MWHS-2 observations are used, we have added the following text to Sect. 2.1.1:
    Pg 4, line 104: *"For the demonstration of the study, MWHS-2 simulations from the ECMWF model background are used. Actual measurements are not taken into account. The requisite data was obtained from ECMWF. More details are described in Sect.2.1.1."*

- L137 "Simulations for all three sensors are noise free, so to incorporate the measurement uncertainties, whenever needed, Gaussian noise is added according to the channel NEDT (Table 1 - Table 3)." Are the errors arising from the radiative transfer calculation accounted for?

    Reply: The most important errors arising from radiative transfer calculations, are related to representing the cloud microphysics. In this study, for ICI and SMS, we consider only one particle size distribution (PSD) and habit and this could underestimate the true cloud variability. Underestimation of scattering at higher frequencies can lead to some imperfections in mapping the cloud information from sub-mm and $183\,\mathrm{GHz}$. Other factors affecting the accuracy of simulations, but not considered due to brevity include neglected antenna pattern and limitations associated with input data, both Cloudsat and ERAInterim. For

example, the simulations could have tendency to be biased towards the Cloudsat geographical sampling. The actual background departures and the corresponding bias correction shall only be revealed when data from ICI is available in future.
We have added a short description these radiative transfer errors that could affect the simulations in Sect. 2.2.

- L159: "for all selected quantile fractions " by quantile fraction, do you mean the n th amongst the 7 selected (from 0.2 to 99.8%)? Also, 16, 50 and 85% are not symmetric (rounding?)

  Reply: Yes, the selected quantiles are the seven percentiles mentioned. We re-phrase some sentences here to make it more clear. 85% is a typo, it should be 84%. It has been corrected.

- L163: Pfreundschuh uses an indicator function I (=1 or 0) in the CRPS, the authors here use y, can they explain the difference?

  Reply: The CRPS in the manuscript is incorrect. We thank the reviewer for highlighting it. We have corrected the equation in the revised manuscript.

- L173: " The input data is all-sky brightness temperature" the simulated one, even for MWHS2, right?

  Reply: Yes, even for MWHS-2 we use simulated all-sky brightness temperatures.

- L301: "0.15 K" should be -0.15

  Reply: The correction is made.

- L453: "Among these four channels, 150 GHz has the highest peaking function around 4 C2km (Chen and Bennartz, 2020)" Could you please clarify what you mean here, 150GHz is neither the highest nor the lowest peaking channel nor it peaks at 4km. Channel 89, 118+/-1.1, 118+/-2.5, and 150 GHz peak at 0.1, 9.6, 2.9, 1km, respectively (according to Chen and Bennartz, 2020), this can also be seen in Lawrence et al. (2018) Fig. 1 through the Jacobians of channels 6 and 7.

  Reply: The reviewer is correct. It was wrongly mentioned that 150 GHz is the highest peaking channel, at $4 \, \text{km}$. We have corrected the text and it now reads as:
  Pg 25, line 465: *"The channels, 150 GHz and 118.75±2.5 peak between surface and 4 km. These channels can only provide coverage to the humidity channels in the lower and mid troposphere. However, the 183 GHz channels are sensitive to hydrometeor content up to 10 km. The channel 118.75±1.1 peaks around 10 km, but such information is only partly relevant for the higher peaking channels of 183 GHz."*

- L460: "but such information would be not be completely orthogonal" duplicate "be"

Reply: The duplicate word is removed.

- Table 1: NEDT are constructor specifications, the real noise is lower, see Fig 5 Guo, Y., J. Y. He, S. Y. Gu, and N. M. Lu, 2019: Calibration and validation of Feng Yun-3-D microwave humidity sounder II.
  IEEE Geoscience and Remote Sensing Letters, doi:10.1109/LGRS.2019.2957403. Tab 5 Carminati, F., Atkinson, N., Candy, B., Lu, Q.: Insights into the Microwave Instruments Onboard the Feng-Yun 3D Satellite: Data Quality and Assimilation in the Met Office NWP System. Adv. Atmos. Sci. (2020). https://doi.org/10.1007/s00376-020-0010-1

  Reply: The reviewer rightly mentions that the real noise is lower, however we add the sensor noise according to the pre-launch specifications. This can be viewed as a conservative estimate of the sensor noise to include other sources of error not accounted for in NEDT. For example in the study by Carminati et al. (2020) the authors describe radiation leak affecting the higher frequency channels.
  We clarify in the manuscript that pre-launch specification values are used.

**Questions**

- Is this method applicable to IR e.g. to an ATOVS system?

  Reply: Yes, the method could potentially be applied to IR. The operation and performance of the cloud correction approach is based on the fact that different frequencies have varying sensitivity to cloud signatures in the same field of view. For IR, infact this feature is regularly used to flag out cloud contamination. Most cloud flagging schemes for IR are based on brightness temperature thresholds.
  In the manuscript, without going into the details, we mention that "*The scheme could also be potentially extended to cloud correction at infra-red frequencies*"

- The authors explain that it could benefit the all-sky assimilation systems indirectly for the analysis increment. Instead (or in addition) could it be used to model the variable observation error (when and by how much to be inflated)? This would be, in my view, the most valuable.

  Reply: The reviewer has a very good point. At ECMWF, the observational errors for MHS and MWHS-2 are defined as quadratic functions of symmetric cloud indicator (Geer et al., 2014). The observational errors are higher in regions for cloud and vice-versa. Thus it could be potentially feasible to use the QRNN identified cloud impact to formulate the observational errors. This could probably be the best use of the QRNN technique for all-sky. Of course using QRNN to develop a full observational model, would require to characterize the performance of the scheme over upper latitudes, land and ocean.
  We have included the following text in the manuscript:
  Pg 29, Line 577: "*Also, it could be feasible to use the QRNN identified cloud impact to formulate the observational errors. This may be the best use of the QRNN technique when it comes to all-sky assimilation.*"

- Could the uncertainty use for weak constraint 4dvar?

  Reply: Although, the uncertainties obtained from QRNN do not have a direct application to weak constraint 4dvar, but they could still be considered as a diagnostic to help evaluate the weak constraint in the troposphere. For example, undiagnosed cirrus contamination in the upper troposphere might be associated with apparent model biases that are actually caused by systematic forward model errors.

- What is the resource cost of this method (is this fast enough to be used in 1-h regional nwp with 30min window)?

  Reply: The method is probably computationally more complex than other methods based on, for example, a scattering index, we consider it unlikely that the computational cost of the scheme would be an issue in an operational context.
  The only demanding part of the scheme is the training process, but it is performed offline. Evaluating the network during operational processing would require only a forward pass through the network. In the study, we employ a simple, fully-connected network architecture so that the complexity of a forward pass is dominated by a low number of matrix-vector multiplications (one for each layer the network). The matrix-vector multiplications are typically combined into matrix-matrix multiplications to evaluate the network for multiple observations in parallel. Due to the recent popularity of neural networks, highly optimized implementations of these methods are available for all common computing architectures. We have included the following information in the Conclusions section. Pg 29, Line 591: *"Due to its low computational cost, implementation of this scheme should be feasible in NWP models given their computational constraints. Although the method is probably computationally more complex than existing cloud clearing methods, the demanding part of the scheme, the training, is performed offline. The operational processing only requires a forward pass through the neural network, for which highly-optimized implementations are readily available on all common computing platforms."*

**References**

Carminati, F., Atkinson, N., and Candy, B.: Insights into the Microwave Instruments Onboard the Feng-Yun 3D Satellite: Data Quality and Assimilation in the Met Office NWP System., Adv. Atmos. Sci., https://doi.org/https://doi.org/10.1007/s00376-020-0010-1, 2020.

Geer, A. J., Baordo, F., Bormann, N., and English, S.: All-sky assimilation of microwave humidity sounders, Tech. Rep. 741, ECMWF, https://doi.org/10.21957/obosmx154, URL https://www.ecmwf.int/node/9507, last access: :29 October 2020, 2014.

---

## Author Comment (AC2) · 4 Mar 2021

**Response to Reviewer 2**

The authors would like to thank the reviewer for his/her careful review and constructive comments, which we believe will help improve the content of the manuscript. Below, under each bullet point, we provide a point by point response to each comment/question. The author responses are given in blue, and textual changes are italicised.

**General Comments**

- As I understand it, this study uses only simulated measurements (either with or without hydrometeors included), even for MWHS-2 for which actual measurements are available. This should be made clearer, especially given line 66, which states that no "background" data is required for the method. I'd also like to see a bit more discussion about whether the results you present should be expected to hold for real data. An assumption that you are making is that your forward model can represent real clouds with enough fidelity that a model trained on simulated data will still work on real-world data. Can you point to any evidence to back up this assumption? Have you tried comparing the histograms of model-simulated Tbs with real-world MWHS-2 Tbs?

  Reply: We agree with the reviewer, the line 66, can be confusing. We have re-written the sentence to:
  Pg 3, line 68: *"This is done for each channel separately and by only using the measurements, although the scheme is demonstrated in the study by using simulated observations."*

  Regarding the discussion about simulations holding well for the real-data, we agree with the reviewer that our assumption is a strong one and requires some evidence. For MWHS-2, we have added a new figure (Pg 6, Fig. 1, also shown below) comparing the histograms of background and bias corrected observations for one of the lower peaking channels of 183 GHz channel. The main deviations in the distributions arise from the hydrometeor scattering. With limited scope of particle size and shape variation in current NWP microphysical schemes, the true cloud variability in radiance space is likely underestimated, though this is but one factor among many when it comes to the challenge of modelling clouds. Also, ECMWF has real time monitoring of many satellite sensors including FY-3D MWHS-2, thus it is possible to assess how well the system (and RTTOV-SCATT) is performing.
  For ICI and SMS, though a qualitative assessment of the simulations is not possible, we expect underestimation of the cloud variability owing to only single PSD and habit assumption. However, it is important to consider that QRNN can easily adapt to changes in brightness temperatures introduced by PSD and habit variation.
  We have added this discussion in the revised manuscript in Sect 2.2.

- What is the computational cost of the QRNN method compared to simpler cloud- clearing filters? Could it feasibly be implemented into existing NWP models given current compu-

[Figure]

Figure 1: Probability distribution functions (PDFs) of simulated and observed brightness temperatures for MWHS-2 channel 14. The data covers latitude range $60°$ S to $60°$ N, and satellite zenith angle less than $7.5°$.

tational constraints?

Reply: Although the method is probably computationally more complex than other methods based on, for example, a scattering index, we consider it unlikely that the computational cost of the scheme would be an issue in an operational context.
The only demanding part of the scheme is the training process, but it is performed offline. Evaluating the network during operational processing would require only a forward pass through the network. In the study, we employ a simple, fully-connected network architecture so that the complexity of a forward pass is dominated by a low number of matrix-vector multiplications (one for each layer the network). The matrix-vector multiplications are typically combined into matrix-matrix multiplications to evaluate the network for multiple observations in parallel. Due to the recent popularity of neural networks, highly optimized implementations of these methods are available for all common computing architectures. We also include the following text to the Conclusions.
Pg 29, Line 591: "*Due to its low computational cost, implementation of this scheme should be feasible in NWP models given their computational constraints. Although the method is probably computationally more complex than existing cloud clearing methods, the demanding part of the scheme, the training, is performed offline. The operational processing only requires a forward pass through the neural network, for which highly-optimized implementations are readily available on all common computing platforms.*"

- Lines 135-143: Why the large difference in number of cases simulated for ICI ($220\,000$) vs. SMS ($143\,000$), if they are both coming from the same population of CloudSat profiles? For the training dataset, you use about 75% of total cases for MWHS-2, about 80% of cases for ICI, and about 84% of cases for SMS. Why this difference in proportions?

Reply: Yes, both ICI and SMS simulations come from same population of Cloudsat. For

SMS, we had access to a smaller database, which was used for a earlier study. Additionally, SMS has fewer channels involved so it can be handled by smaller databases.

Due to difference in sizes of the available observations, maximum possible observations were incorporated in the training data to allow cloud variability. Even with differences in the database sizes, the results show that the predicted quantiles are well calibrated for all three sensors. With a larger database, one could expect similar or even better performance.

- Line 340: "With a test dataset of 70 000 samples, we cannot represent the far wings of the distribution accurately." Couldn't one apply this same reasoning to Figures 2,3, or 8, which have density values below $10^{-4}$ included?

  Reply: Yes one could apply the same reasoning to the Figures 2, 3, or 8, but we prefer to show the complete distribution to emphasize that the cases with large errors occur infrequently.

- Figure 13: I wonder if there might be an easier, more concise way to evaluate whether the uncertainty intervals are properly calibrated. Namely, could you simply calculate how often the true value falls within the $\pm 2\sigma$ uncertainty range, for each of the uncertainty bins $(0 - 3K, 3 - 8K, 8+ K)$ that you've included on the plot? If this percentage is significantly less than 95%, it would suggests that your uncertainties are too small, while if it were closer to 100% it would suggest your uncertainties are too large. Even if you don't get rid of Fig. 13, I think this would be useful information to include in the paper.

  Reply: As per suggestion of the reviewer, we have included the information highlighting how well calibrated the uncertainties are. We provide the occurence percentage of true value for each uncertainty bin and add this information in Fig. 13 (also shown below, as Fig.2). For each bin considered, the true value falls in the $\pm 2\sigma$ uncertainty range almost 94% of the time, indicating that the uncertainties are well calibrated. Only for channel I3V and the outermost bin, occcurence percentage is 88%.

  These results are in line with Fig.7 and Fig.11, which also indicate that the uncertainty intervals are well calibrated.

- Lines 498-499: ". . . other underlying uncertainties not considered here." Whether here or elsewhere in the paper, I think you should talk a bit more about what these other uncertainties are (radiative transfer model errors should certainly be discussed), and what effects they might have on your results.

  Reply: Among the errors which affect the radiative transfer calculations are underestimation of cloud variability due to limited particle size distribution (PSD) and habit variation. Other factors include neglected antenna pattern and the limitations associated with input data, both Cloudsat and ERAInterim. For example, the simulations could have tendency to be biased towards the Cloudsat geographical sampling. All these issues can contribute to mapping errors between 183 GHz and higher frequency channels. However, it is possible improve the cloud representation by incorporating PSD and habit variation, and QRNN based cloud correction can easily adapt to changes in brightness temperatures introduced by the local variability. This is not the case with all-sky data assimilation systems, which may assume only single PSD and habit combination.

[Figure]

Figure 2: Distribution of errors binned according to their uncertainty. The percentage values in the parenthesis represent the occurence of true value within the $\pm 2\sigma$ uncertainty range, for each of the uncertainty bins. Results are from QRNN-single for channels I1V, I2V and I3V.

We have edited Sect 2.2 and Conclusions to describe these different errors and their implications.

**Typos**

- Line 26: The phrase "weather satellites are since some time equipped" is confusing to me. I suggest "weather satellites have for some time been equipped. . ."

    Reply: The sentence is re-written as per suggestion.

- Line 51: I believe you are missing the word "on" between "predicated" and "Gaussian."

    Reply: The typo is corrected.

- Line 154: Should 85% be changed to 84%? That would be +2 sigma for a normal distribution.

    Reply: Yes, the predicted percentile should be 84% instead of 85%. The typo is corrected.

- Line 455:". . . provide coverage to the humidity channels in, lower and mid troposphere" This is confusing - get rid of the comma and add the word "the" instead?

    Reply: The comma is removed and the word "the" is added.

- Line 494: "QRNN predictions are weighted mean. . ." I think this should say, "QRNN predictions are the weighted mean. . ."

    Reply: The word "the" is added.

- Line 508: Missing "the" before "same"

    Reply: The word "the" is added.